# The association between socioeconomic status and pandemic influenza: Systematic review and meta-analysis

**Svenn-Erik Mamelund** [1]* , **Clare Shelley-Egan** [2] , **Ole Rogeberg** [3]

**1** Centre for Research on Pandemics & Society, Oslo Metropolitan University, Oslo, Norway, **2** Work Research Institute, Oslo Metropolitan University, Oslo, Norway, **3** Frisch Centre, University of Oslo, Oslo, Norway

☯ These authors contributed equally to this work.

* masv@oslomet.no

## Abstract

### Background

The objective of this study is to document whether and to what extent there is an association between socioeconomic status (SES) and disease outcomes in the last five influenza pandemics.

### Methods/principle findings

The review included studies published in English, Danish, Norwegian and Swedish. Records were identified through systematic literature searches in six databases. We summarized results narratively and through meta-analytic strategies. Only studies for the 1918 and 2009 pandemics were identified. Of 14 studies on the 2009 pandemic including data on both medical and social risk factors, after controlling for medical risk factors 8 demonstrated independent impact of SES. In the random effect analysis of 46 estimates from 35 studies we found a pooled mean odds ratio of 1.4 (95% CI: 1.2–1.7, p < 0.001), comparing the lowest to the highest SES, but with substantial effect heterogeneity across studies,–reflecting differences in outcome measures and definitions of case and control samples. Analyses by pandemic period (1918 or 2009) and by level of SES measure (individual or ecological) indicated no differences along these dimensions. Studies using healthy controls tended to document that low SES was associated with worse influenza outcome, and studies using infected controls find low SES associated with more severe outcomes. A few studies compared severe outcomes (ICU or death) to hospital admissions but these did not find significant SES associations in any direction. Studies with more unusual comparisons (e.g., pandemic vs seasonal influenza, seasonal influenza vs other patient groups) reported no or negative non-significant associations.

### Conclusions/significance

We found that SES was significantly associated with pandemic influenza outcomes with people of lower SES having the highest disease burden in both 1918 and 2009. To prepare

**Data Availability Statement:** All relevant data are within the manuscript and its S1 and S2 Files and S1 Checklist and S1 and S2 Tables files.

**Funding:** This research is part of the project PANRISK: Socioeconomic risk groups, vaccination and pandemic influenza, funded by a research grant from the Research Council of Norway (grant agreement No. 302336).

**Competing interests:** The authors have declared that no competing interests exist.

for future pandemics, we must consider social vulnerability. The protocol for this study has been registered in PROSPERO (ref. no 87922) and has been published Mamelund et al. (2019).

## Introduction

It used to be believed that pandemic and infectious disease risks are the same for all, irrespective of socioeconomic status (SES). But when 61-year old superstar Madonna shared this belief on Instagram on the 23$^{rd}$ of March 2020, calling COVID-19 "the great equalizer" from a milky bath sprinkled with rose-petals [1], fans and others quickly pointed to the disproportionate pandemic burden and suffering of the poor. Indeed, their criticism is supported by a number of studies showing that certain indigenous people, people of colour, immigrants and the poor have experienced disproportionate harm as a result of COVID-19 as measured by infection rates, hospitalizations, the need for intensive care unit treatment, and death [2–5].

The idea that outcomes from infectious disease pandemics are socially neutral has a long history among lay people, researchers and policy makers responsible for pandemic preparedness plans. Literature on SES and 1918 influenza outcomes published by social historians between 1970 and 1990 argued that the disease was so highly transmissible that everybody was equally affected [6–10], pointing to anecdotal evidence such as the president and King of Spain falling ill and the Swedish Prince Erik dying at age 29 [11]. However, these studies used aggregate-level data, were mainly descriptive and did not use multivariate statistical models. Empirical studies appearing from the mid-2000s often reported evidence inconsistent with the socially neutral hypothesis: SES seemed to be linked to exposure, susceptibility and access to care, and SES indicators were statistically associated with mortality [12–14]. Although several studies of the 2009 pandemic also found SES associated with various pandemic outcomes [15–17], this social inequality in risk is still ignored in international influenza pandemic preparedness plans [18]. Apart from a systematic review and meta-analysis of the 2009 pandemic disease burden in low and low to middle income economies and differences in disease outcomes in that pandemic for ethnic minorities vs non-ethnic minorities [19], a systematic assessment of several influenza pandemics and of the evidence for disparities in pandemic outcomes by individual and/or area-level SES (e.g. education, income, household crowding and quality, unemployment, occupation-based social class, poverty status, share below poverty levels, deprivation indexes etc.) has been lacking.

In this paper, we present the first systematic review and meta-analysis on the association between SES and disease outcomes in the last 5 influenza pandemics. The objective is to document whether and to what extent there is an association between indicators of socioeconomic status (e.g. income, education) and pandemic outcomes (infection, hospitalizations, mortality) in the last five influenza pandemics (1889, 1918, 1957, 1968, 2009). In terms of PICOS criteria, the Population (P) consists of groups defined by socioeconomic status, the intervention (I) or exposure or risk factor is pandemic influenza, the comparison (C) or alternative interventions is not relevant, while the outcomes (O) are morbidity, hospitalization, or death associated with influenza pandemics. All types of study design were considered (S). As described in our pre-registered analysis plan, we hypothesized that the association between SES and pandemic outcomes would increase with outcome severity, as higher income and SES tend to be associated with access to resources and protective factors that reduce the risk of progression to more severe outcomes.

## Materials and methods

### Bibliographic database search

A systematic search of Medline, Embase, Cinahl, SocIndex, Scopus and Web of Science was performed to identify all relevant articles published on SES and pandemic influenza (morbidity, severe disease and mortality). SES was captured by keywords such as education, income, occupational social class etc. (see search strategy, S1 Table, for more examples). Morbidity was captured by keywords such as infection rates, transmission rates, lab confirmed influenza, flu like illness, and influenza like illness (ILI). Severe disease was captured by keywords such as disease severity, critical illness, critical disease, severe illness, severe disease, hospitalization, patient admission, hospital admission, intensive care unit (ICU) admission, and ICU treatment. Mortality was captured by keywords such as fatal outcome, fatal illness, fatal disease, fatality, lethal outcome, lethal illness, lethal disease, terminal outcome, terminal illness, terminal disease, lethality, death, death rate, and mortality rate. All of these keywords were used in both pilots and the final searches. The strategy for the literature search was developed by two information specialists in cooperation with the research group, starting 5 October 2017. Several pilot searches were conducted in Web of Science and Medline respectively, on 12 and 19 October 2017, to ensure a sensitive search. The search strategy combined relevant terms, both controlled vocabulary terms (i.e. MeSH) and text words. The main search strategy used in Medline is available in PROSPERO 87922 and in S1 Table. The final search was carried out on 17 November 2017. The strategy was modified to fit the other databases listed above. To generate manageable results, restrictions on language (English, Danish, Norwegian and Swedish) and publication type (article/research article) were added to the searches in the other databases. The searches in Medline and Embase were performed without publication type restrictions. The search strategy was peer-reviewed by a third information specialist using a structured tool based on the PRESS-framework [20]. Reference lists of relevant known studies were also screened and experts in the field consulted in order to identify other additional sources. Finally, we also contacted authors of published studies to ask for relevant data not presented in the papers or in appendices. However, we did not get any responses to these requests.

### Inclusion criteria for title and abstract screening

After adding all identified records to an Endnote library and removing duplicates, the remaining results were imported to the program Covidence. Here, additional duplicates were removed. The title and abstract of each article was screened by two of the authors (SEM and CSE), according to the selection criteria. After screening of titles and abstracts, we added full-text versions of articles in Covidence. Divergences in the inclusion of studies were re-assessed by the same researchers until consensus was reached in terms of inclusion or exclusion. The criteria for inclusion were:

1. The study period 1889–2009 includes the five pandemics in 1889, 1918, 1957, 1968 and 2009

2. Studies investigating the association between SES and pandemic outcomes

3. Studies of race, ethnicity, and indigenous people that reported data on SES controls

4. Studies addressing both seasonal and pandemic influenza distinguishing between non-pandemic and pandemic years

5. Studies addressing all regions/countries, type of studies (interventional, observational, etc.) and populations (age, gender, pregnant women, soldiers etc.)

### Exclusion criteria for title and abstract screening

The following criteria excluded studies from the systematic literature review:

1. Studies on pandemic diseases other than influenza

2. Studies on seasonal influenza only

3. Studies on both seasonal and pandemic influenza that *did not* distinguish between non-pandemic and pandemic years

4. Studies on attitudes and compliance with (non)pharmaceutical pandemic interventions

5. Qualitative studies on the associations between SES and pandemic outcomes

6. Studies on social justice and pandemic influenza

7. Studies of pandemic influenza preparedness plans

8. Studies of race, ethnicity, and indigenous people that *did not* report data on SES controls

### Data selection and extraction

We drafted a data abstraction form, pilot-tested it and modified it, when necessary. Two reviewers (SEM and CSE) independently extracted data from all included studies. Any disagreements were resolved via discussion or by involving a third reviewer for arbitration. 1–5 and 6 below were entered into separate spreadsheets for each article. The following information was extracted:

1. Article info

    a. First author

    b. Year published

    c. Journal

2. Data sample

    a. Country or region of analysis

    b. Pandemic years (1889, 1918, 1957, 1968, 2009)

    c. Sample inclusion criteria–i.e. characteristics of sample/population (civilian, military, gender, pregnant, age-group/median/average age, patient group etc).

    d. Sample size

    e. Unit of analysis (individuals, households, regions, hospitals etc)

    f. Data aggregation level (observations of individual units, aggregated units, etc.). e.g., if hospitals are the unit of analysis, does the data used occur at the hospital level or is it pooled across hospitals?

    g. Source of outcome data, e.g., census, routine notification data (e.g. influenza cases reported to a doctor), survey data, register data

        i. If survey or population data had incomplete coverage

            1. Response rate/coverage

2. Representativity: Is the sample shown to be representative for the population? i.e. has a non-response analysis been carried out?

3. Outcome variable—Pandemic outcome (a. morbidity, b. hospitalization, c. mortality)

   a. Definition of morbidity: influenza-like illness (ILI), Lab-confirmed Infection rates (PCR), transmission rates (reproduction number, R0), immunity/antibodies towards influenza (HI titer above a certain threshold) due to exposure to the disease and not vaccination

   b. Definition of hospitalization; Hospitalized inpatients with (PCR) or without confirmed influenza; patients admitted to intensive care unit (ICU) or not; mechanically ventilated patients ("lung machines") or not; inpatients vs outpatients

   c. Definition of cause of mortality: Influenza and pneumonia (PI), excess mortality (PI, all causes of death etc.), respiratory diseases, pneumonia etc.

4. Baseline outcomes (control type), i.e. what was the control group or baseline outcome comparison? (general healthy population, infected patients, the hospitalized, patients with lab-confirmed seasonal influenza)

5. Independent variables of interest–relating to SES

   a. Type of SES indices (education, income, crowding, density, deprivation index, unemployment, occupational social class, poverty status, % below poverty level)

   b. Definition or brief descriptive text on SES indices (e.g., if based on a specific type of poverty index etc.)

6. Statistical methodology

   a. Design of study (cross sectional, longitudinal, case-control, cohort studies)

   b. Estimation technique (Cross tables, correlation analysis, OLS, Poisson regression, Logistic regression, Cox regressions, GEE regressions, GLMM models etc.)

   c. Control variables included (e.g. age, gender, marital status, pre-existing disease, health behavior etc.) in light of sample restrictions (e.g. for pregnant women, sex is not among the controls)

   d. Reference categories with which all point estimates are compared

7. Results reported (separate spreadsheet)

**Data synthesis.** Our narrative review includes a table of the study characteristics of the included studies, such as study authors and year, pandemic years, study region (region/country/hospital), sample inclusion criteria, sample size, unit of outcomes, data aggregation level, data sources and type, outcomes, baseline outcomes, SES measure, design, statistical techniques, controls, whether the study estimates are used in the meta analysis, and whether SES is an independent predictor. The quantitative part of the study pools results across individual studies using meta-analytic methods.

In the simplest meta-analytic model ("fixed effect") random sampling variation is assumed to be the only source of variation in estimates. This is implausible in our context, in which studies use different SES indicators and medical outcomes from different countries and time periods. A "random effect" model captures the resulting *effect heterogeneity* by estimating the distribution of these underlying associations. Systematic variation across study-level covariates

(e.g., pandemic period, region, type of outcome) is assessed using sub-sample analyses as well as a Bayesian hierarchical model.

We searched the identified studies in our meta-review for quantitative estimates of associations between SES indicators and influenza related outcomes. The resulting estimates were assessed for inclusion in the meta-analysis, and included if they could be expressed as an odds-ratio or relative risk for low versus high socioeconomic status. This implied that estimates had to include an indicator of socioeconomic status at the individual or ecological level, and had to allow for an estimate of how the incidence or prevalence of some flu related outcome varied by levels of this indicator. Where studies included estimates for distinct data subsamples (different age groups, periods), single estimates pooling all data were preferred if available. If not, the separate estimates were all included. For some studies, multiple estimates were also extracted if they performed different comparisons (e.g., risk of infection, and risk of hospitalization given infection). We also collected study level factors indicating the pandemic period (1918 vs 2009), country/region, and data as to whether the study estimate involved an odds ratio or a relative risk or rate. The specific studies included and all judgments and adjustments concerning inclusion and adjustments of reported numbers are detailed in the S2 File.

We have been inspired by NOS [21] to assess the quality of the included studies. We have rated the following items: A) *Selection* of exposed population (Broad and representative sample/population of the exposed?, truly and somewhat = 1, selected sub-groups and no description of population = 0) and non-exposed populations (1 = drawn from same community as the exposed, 0 = drawn from different source or no description); B) *Comparability* (confounder controls (yes = 1, no = 0), biological controls (yes = 1, no = 0), SES measure significant beyond biological controls (yes = 1, no = 0); C) *Data quality* (Lab-confirmed outcomes (yes = 1, no = 0) and data aggregation level (individual = 1, aggregate = 0)); The average and median quality score in the 44 studies included in the narrative review is respectively 4.5 and 4, while min score is 2 and max 7. The quality scores were higher in the 35 studies included in the meta-analysis (average 4.9, median 5, min 3, max 7) than those 9 not included (average 3.1, median 3, min 2, max 4) (see the quality assessment scores per article in the S2 Table).

Relative to the pre-analysis plan, the ambitions of the quantitative analysis were scaled back given the large heterogeneity across the studies included (see Table 1). The pre-analysis plan specified three types of analysis [22]. The first, a standard random effect analysis with subsample analyses, was conducted as planned using the «REML» algorithm in the Metafor meta-analytic package for R [23]. The second, a PET-PEESE analysis testing and adjusting for publication bias, was found unsuitable given the large effect heterogeneity [24]. The third, a Bayesian model to assess "dose-response" effects and assess how estimates vary with study-level indicators and the type of comparisons made, is included in a simplified version without the "dose-response" element.

The Bayesian model differs from the standard fixed and random effect models in two ways. It allows us to include study-level covariates to capture systematically different effects in specific regions, periods or for specific outcomes, using a hierarchical specification across the parameters to impose partial pooling and reduce the risk of large but spurious estimates [25]. If the evidence as a whole indicates that estimates vary no more across study level indicators than we would expect due to sampling variation, then this will pull the individual indicator coefficients towards zero.

Second, the Bayesian model requires a prior distribution for each model parameter that expresses reasonable (pre-analysis) beliefs regarding the parameter values. The estimation updates these beliefs in light of the data, resulting in a posterior distribution that blends the pre-existing knowledge encoded in the prior distribution with evidence from the observed data. To verify that the prior choices for the overall pooled effect and heterogeneity do not

**Table 1. Overview of 44 studies included in the systematic review by study characteristics.**

| * | Study authors and year | Study region | Pandemic period | Sample inclusion criteria | Sample size | Unit of outcomes | Data aggregation level | Data sources | Outcomes | Baseline outcomes | SES measures | Design | Statistical technique | Controls | Estimates used in meta analysis and is SES an independent predictor? |
|---|---|---|---|---|---|---|---|---|---|---|---|---|---|---|---|
| 2 | [32] | London, England | 20 April- 28 June 2009 | People of all ages seeing a doctor for influenza at hospitals and community clinics in London | 2,819 H1N1 patients (confirmed, presumed and probable) with valid LSOA postcodes | Individuals | Individual cases, but SES of cases based on the IMD of area post-codes | Data on cases and contacts were from the London Flu Response Center database and where coupled to IMD 2007 | Influenza cases per 100,000 | Population at risk in each LSOA area | Area Index of multiple Deprivation (IMD) 2007 quintiles (economic, social and housing issues) | Cross-sectional univariate design | Bivariate rate ratios with 95% CI | Age and weekly interactions with IMD | Meta analysis: Yes (all ages and whole period) SES measure significant |
| 3 | [33] | New York, USA | 24 April-7 July 2009 | Active hospitalized-based surveillance and passive collection of on demographics, risk conditions, and clinical severity | 996 H1N1 patients (929 Confirmed and 67 probable) | Individuals | Individual cases, but SES of cases based on United Hospital Fund Poverty neighborhoods | Active hospitalized-based surveillance and passive collection of on demographic, risk conditions, and clinical severity | Hospitalizations per 100,000 | Population at risk in high, medium or low poverty areas | Tertiles of percentage of residents living <200% of the federal poverty level according to the 2000 US Census | Cross-sectional univariate design | Bivariate Rate ratios with 95% CI | Age | Meta analysis: Yes SES measure significant |
| 4 | [34] | New Zealand | Nov 2009-March 2010 | Randomly selected serum samples from GPs countrywide and in the Auckland region 3 months after the pandemic | 1,687 serum samples | Individuals | Individual observations | seroprevalence data coupled with questionnaires evaluating demographics and potential risk factors. | H1N1 Infection rates (Seroprevalence; Antibody titer >1:40) | Baseline immunity was measured from 521 sera collected during 2004 to April-2009 | Damp housing (poor housing conditions is an often used measure of SES, see [67]) | Multi-stage random cross-sectional design | Multivariate logistic regressions | Age, ethnicity, gender, vaccination history, chronic illness | Meta analysis: Yes SES measure ns. |
| 6 | [35] | Eight cities in Hamedan Province, western Iran | July-December 2009 | Subjects (cases and controls) were selected from patients with signs and symptoms of respiratory tract infection who were referred to health centers | 245 cases and 388 controls | Individuals | Individual observations | Data are from health centers on H1N1 infection status coupled with covariate data from interviewers using predetermined questionnaires | Cases were identified by pharyngeal soap specimens positive for influenza A virus using PCR | Controls were testing negative for influenza A virus using PCR | Education 1. low education: illiterate, primary school and middle school. 2. High education: high school and academic | Unmatched case-control study | Multivariate logistic regressions | Age, sex, pregnancy, suspected close contact with influenza patients, smoking, region (urban-rural), trip during last week, chronic disease, influenza vaccination, and BMI | Meta analysis: Yes SES measure significant, but unexpectedly higher risk for the high education group. |
| 7 | [26] | England & Wales | 12 Oct 1918-5 April 1919 | Influenza deaths in all parts of E & W | - | Aggregate: 305 adm. units & 62 counties | Aggregate | Deaths from National vital registration systems and demographic data from the 1921 census | Influenza death rates and reproduction number R (the average number of secondary cases generated by an index case) | Population at risk | People per acres, dwellings and rooms | Cross- Cross-sectional control-variable design | Spearman correlations, using a Bonferroni correction for multiple comparisons (transmissibility and death rates) and multivariate logistic regressions (death rates) | Population size, fall and winter waves, urban-rural | Meta analysis: No There were no association between transmissibility, death rates and indicators of population density and residential crowding |
| 10 | [36] | Global (226 studies from 50 countries met the inclusion criteria) | 2009–2010 | Described confirmed, probable or suspected cases of 2009–2010 influenza A (H1N1) infection; and (2) described patient(s) who were critically ill | 10695 | Individuals | Aggregate, Global | Medline, Embase, LiLACs and African Index Medicus to June 2009-March 2016 | Mortality associated with H1N1-related critical illness | Population at risk | World Bank economic development status of countries (High, upper middle, lower middle income) | Systematic review and meta analysis | Random effects meta regressions | No controls | Meta analysis: No SES measure significant |

*(Continued)*

**Table 1.** (Continued)

| * | Study authors and year | Study region | Pandemic period | Sample inclusion criteria | Sample size | Unit of outcomes | Data aggregation level | Data sources | Outcomes | Baseline outcomes | SES measures | Design | Statistical technique | Controls | Estimates used in meta analysis and is SES an independent predictor? |
|---|---|---|---|---|---|---|---|---|---|---|---|---|---|---|---|
| 11 | [37] | Mexico | 10 April to 13 July 2009 | Data from clinical files from all influenza A deaths | 239 H1N1 cases and 85 influenza A controls | Individuals | Individual observations | Patients' clinical records and reporting forms from health facilities | Lab-confirmed A/H1N1 deaths (rt-PCR-test) | Seasonal influenza A deaths | Education (Primary school or less, junior high school, High school or higher level) | Case-control | Propensity score multivariate logistic regressions | Sex, age, have a partner, smoking, employment status | Meta analysis: Yes SES measure ns. |
| 12 | [38] | Canada (Quebec) | 16 April-1 July 2009 | Lab-confirmed H1N1 hospitalizations or ICU admission/ deaths | 321 hospitalized incl. 47 ICU and 15 deaths (cases) and 395 non-hospitalized N1H1 infection patients (controls) | Individuals | Individual observations | Suspected H1N1 case at primary care clinics or hospital coupled with other data from standardized questionaries' | Lab-confirmed influenza associated hospitalizations (24 hrs or more) and ICU/death | Non-hospitalized H1N1 patients (vs. hospitalized non-severe (vs. ICU/death) | Education (high school not competed, non-University certificate, university degree) | Case-control | Multivariate logistic regressions | Age, sex, HCW, smoking, flu jab in 2008–09, consultation, days after onset, antiviral use, pregnancy, underlying condition, obesity | Meta analysis: Yes (both outcomes included) SES measure ns. |
| 13 | [39] | Spain (Andalusia, Basque Country, Catalonia, Castile and Leon, Madrid, Navarra and Valencia) | July 2009-Febr. 2010 | Lab-confirmed hospitalization (RT-PCR) | 699 hospitalized and 703 non-hospitalized cases of a (H1N1) infection | Individuals | Individual observations | Data from 36 hospitals and primary care centers in 7 spanish regions | Lab-confirmed hospitalizations (patient admitted to hospital for > 24 hours with RT-PCR confirmed H1N1 infection) | Non-hospitalized people with RT-PCR confirmed infection with the same pandemic virus | Education Secondary or higher | Case-control | Multivariate logistic regressions | Age, sex, ethnic group | Meta analysis: Yes SES measure significant. However, data on underlying health collected but not controlled for |
| 14 | [27] | USA (Chicago) | 29 Sep-16 Nov 1918 | Influenza and pneumonia (PI) mortality | 7971 PI deaths | Individuals | Individual deaths, but SES measured at the level of 496 Census tracts | Historical maps of point-level mortality incidence, spatial data and near contemporaneous census data | Influenza and pneumonia mortality and reproduction number (R0) | Population at risk | Census tract-based SES (% illiteracy, unemployment, homeownership, population density) | Cross-sectional control-variable design | Poisson regressions with GEE and Spearman correlations | Age | Meta analysis: Yes % illiterate sig. predictor of mortality controlling for age and all other SES variables. Sig. ass btw. R0 and population density, illiteracy, and unemployment but not homeownership. |
| 15 | [40] | USA (Alaska, Arizona, New Mexico, Oklahoma, Wyoming) | 15 April 2009-31 Jan 2010 | Lab-confirmed A (H1N1) fatalities; state residents who died relating to infection with lab-confirmed influenza A | 145 fatal cases and 236 controls | Individuals | Individual observations | Medical records (notifiable disease reports), death certificates, interviews with cases and controls | Lab-confirmed A(H1N1) fatalities using RT-PCR test | Outpatients with lab-confirmed H1N1 | Healthcare insurance, >1,5 persons per room, graduated high school, poverty (<US$ 25000/year) | Matched case-control | Logistic regressions | Age, sex, race, barriers to health care access, urban-rural, health seeking behavior, vaccination status, health behaviors, pre-existing conditions. | Meta analysis: Yes (poverty) None of the SES variables were significant. |
| 16 | [41] | USA (23 counties) | 23 April-8 June 2009 | English language media reports of A (H1N1) cases | 32 public primary & secondary schools with at least one confirmed H1N1 case and 6815 control schools located in the same 23 counties as the case schools | Aggregate, Schools | Aggregate | Health Map | Media reports of A (H1N1) cases | Schools located in the same 23 counties as the case schools without N1N1 cases | Title 1 school (Whether or not schools qualifies for a federal funding to support economically disadvantaged students. | Matched case-control | Logistic regression | Highest grade at school and size | Meta analysis: Yes SES measure significant |

(Continued)

**Table 1.** (Continued)

| * | Study authors and year | Study region | Pandemic period | Sample inclusion criteria | Sample size | Unit of outcomes | Data aggregation level | Data sources | Outcomes | Baseline outcomes | SES measures | Design | Statistical technique | Controls | Estimates used in meta analysis and is SES an independent predictor? |
|---|---|---|---|---|---|---|---|---|---|---|---|---|---|---|---|
| 17 | [42] | Australia (Brisbane) | Jan-Dec 2009 | Lab-confirmed daily A (H1N1) cases | 11,979 cases | Individuals | Individual cases, but SES measured for postcode areas (SLA) | Queensland Health, SEIFA data from Australian Bureau of Statistics (ABS) & daily rainfall & temperature data from the Australian Bureau of Meteorology | Lab-confirmed daily A (H1N1) cases | Rest of the population with no lab-confirmed case | SEIFA: socioeconomic index for areas, incl. education, occupation and wealth | Cross-sectional control-variable design | Bayesian spatial conditional autoregressive poisson models | Rainfall (mm) and temperature (degrees Celsius) | Meta analysis: No SES measure ns. |
| 18 | [43] | Australia (Queensland) | 7 May-31 Dec 2009 | Lab-confirmed A (H1N1) cases | - | Individuals | Individual cases, but SES measured for postcode areas (SLA) | Queensland Health, SEIFA data from Australian Bureau of Statistics (ABS) & daily rainfall & temperature data from the Australian Bureau of Meteorology | Lab-confirmed daily A (H1N1) cases | Rest of the population with no lab-confirmed case | SEIFA: socioeconomic index for areas, incl. education, occupation and wealth | Cross-sectional control-variable design | Flexible Bayesian, space-time. SIR models | Rainfall (mm) and temperature (degrees Celsius) | Meta analysis: No SES measure significant |
| 19 | [44] | England (West Midlands) | 16 April-6 July 2009 | Lab-confirmed A (H1N1) cases | 3063 cases | Individuals | Individual cases, but SES measured for postcode areas | FluZone, a national surveillance database with case reporting. SES data from IMD 2007 | Lab-confirmed A (H1N1) cases | Rest of the population with no lab-confirmed case | Index of Multiple Deprivation of an area and postcodes (IMD 2007). It includes seven dimensions: income, employment, health deprivation and disability, skills and training, barriers to housing and services, crime and disorder, living environment SES indexes IMD 2007: Index of Multiple Deprivation | Cross-sectional | Descriptive analysis | Age, sex, ethnicity, exposure and illness severity, but no controls were made | Meta analysis: No SES measure significant |
| 21 | [45] | Canada (Rural community of British Columbia; local town and surrounding First Nation reserves | Late April/ early May 2009 | One elementary school and on-reserve aboriginal participants; | 83 ILI cases and 281 non-ILI cases | Individuals | Individual observations | Phone survey of households with at least one child enrolled in any of the community schools | Influenza-like illness (ILI) | Non-ILI cases | Household density | Cross-sectional control-variable design | Generalised linear mixed models (GLMM) | Age, chronic conditions, aboriginal status, received vaccination 2008-09 | Meta analysis: Yes SES measure ns. |
| 23 | [46] | Spain (Andalusia, the Basque Country, Castile and Leon, Catalonia, Madrid, Navarre, and Valencia | July 2009-Feb. 2010 | Patients aged 6 months to 18 years with confirmed H1N1 at 32 Hospitals of the Spanish National Health survey | 195 confirmed H1N1 hospitalized cases and 184 outpatient controls with confirmed H1N1 | Individuals | Individual observations | Spanish National Health Service | Lab-confirmed A (H1N1) inpatient (hospitalized) cases | Outpatient (non-hospitalized) controls with confirmed H1N1 | Parents education (Primary or lower vs. secondary or higher) | Matched case control, prospective, observational study | Logistic regressions | Age, pulmonary, disease, neurological disease, diabetes mellitus, cardiovascular disease, and non-Caucasian ethnicity | Meta analysis: Yes SES measure significant |

*(Continued)*

**Table 1.** (Continued)

| * | Study authors and year | Study region | Pandemic period | Sample inclusion criteria | Sample size | Unit of outcomes | Data aggregation level | Data sources | Outcomes | Baseline outcomes | SES measures | Design | Statistical technique | Controls | Estimates used in meta analysis and is SES an independent predictor? |
|---|---|---|---|---|---|---|---|---|---|---|---|---|---|---|---|
| 24 | [47] | Brazil (Paraná) | 2009 | Patients (in- and outpatients) with lab-confirmed H1N1 infection | 1911 Inpatient cases and 2829 outpatients controlls | Individuals | Individual observations | Brazilian Ministry of Health National Case Registry Database | Lab-confirmed A (H1N1) inpatient cases and outpatient controlls | Lab-confirmed H1N1 outpatients controls | Level of education (Literate vs. illiterate) | Retrospective observational case-control study | Logistic regressions | age, gender, ethnicity, having a comorbidity, number of comorbiditi, 8 types of underlying health conditions, smoking, clinical manifestations, treatment (Oseltamivir), time to treatment initiation in days | Meta analysis: Yes SES measure significant |
| 26 | [48] | USA (New York) | 1 Oct 2009–28 Feb 2010 | Lab-confirmed illness among adults and children | 128 inpatients with lab-confirmed flu cases matched by age and month of diagnosis with 246 non-hospitalized lab-confirmed influenza A controls (assumed to be H1N1) | Individuals | Individual observations | Sentinel surveillance system used by NYC Department of Health and Mental Hygiene; telephone interview to collect clinical and demographic data | Lab-confirmed A (H1N1) inpatient cases and outpatient influenza A controls | Non-hospitalized lab-confirmed influenza A controls (assumed to be H1N1) | Education (Some college or more, not a high school graduate, high school graduate), annual household income and neighbourhood poverty (% Persons living below the federal poverty line) | 1:2 case-control study design, matching by age group and month of diagnosis | Conditional multivariate logistic regressions | Access to care (primary physician, insurance) and at least one underlying condition (various diseases, pregnancy and obesity) | Meta analysis: Yes Education among adults and neighbourhood poverty among children and adults were significant |
| 28 | [49] | Canada (Ontario) | Two waves in 2009 (April 23–July 20 and August 1 Nov 6) | Residents of all ages who received nasopharyngeal swabs and tested positive for H1N1 | 401 self-reported hospitalization cases and 624 non-hospitalized controls (150 hospitalized and 184 non-hospitalized in wave 1, 251 hospitalized and 440 non-hospitalized in wave 2) | Individuals | Individual hospitalizations by individual-level education and contextual level SES variables | Surveillance data and standardised phone interviews | Lab-confirmed A (H1N1) inpatients (hospitalized patients) | Non-hospitalized controls H1N1 positives | individual level education level (of adult participants aged 18 years or older & of parents respondents for children younger than 16 years), household density (individuals per sleeping rooms) and several contextual level SES variables (employment, education, income, social and material deprivation) | Case-control study | Binomial or multinomial logistic regression, using generalized estimating equations to account for clustering/ dependence in the data | Age and gender | Meta analysis: Yes (Total deprivation and individual and parental education for both waves). First wave: High school education or less and living in a neighborhood with high material or total deprivation sign. Second wave: High school education or less sign. Moreover, a mediation analysis showed that clinical risk factors explain only a portion of the ass. btw SES & hospitalization. |
| 29 | [50] | USA (California) | 3 April–15 Sep 2009 | Reported counts of H1N1 hospitalizations, not lab-confirmed | 2010 hospitalizations | 58 counties | Aggregate | California Department of Public Health surveillance data | Reported H1N1 Hospitalizations | Population at risk in each 58 counties | Education (% of persons aged > 25 years with a high school diploma); Poverty (% of pop under poverty line); Income (median HH income in dollars) | Cross-sectional control-variable design | OLS | Sex, race/ethnicity, age, climate, agricultural and transportation variables | Meta analysis: No The 3 SES variables were ns. but results not shown |

(*Continued*)

**Table 1.** (Continued)

| * | Study authors and year | Study region | Pandemic period | Sample inclusion criteria | Sample size | Unit of outcomes | Data aggregation level | Data sources | Outcomes | Baseline outcomes | SES measures | Design | Statistical technique | Controls | Estimates used in meta analysis and is SES an independent predictor? |
|---|---|---|---|---|---|---|---|---|---|---|---|---|---|---|---|
| 30 | [12] | Norway | 1918–1919 | PI deaths covering the whole of Norway | 16,005 deaths | Aggregate, 351 medical districts | Aggregate | Regional district physician reports and census data | PI mortality reported to a doctor | Population at risk | % receiving public support due to poverty; Wealth per person (in 1000 Nok); Average number of persons per room | Cross-sectional control-variable design | OLS | age, sex, ethnicity, % in fishing, coast-inland, summer wave exposure | Meta analysis: No Poverty and wealth, but not crowding was sign. |
| 31 | [13] | Norway (Frogner and Gronland/ Wexels parishes in Oslo) | 1 Feb 1918–1 feb 1919 | PI deaths in the two selected parishes | 250 PI deaths | Individuals | Individuals | Death certificates coupled with census data | PI mortality reported on death certificates | Population at risk | Occupational based social class, apartment size (1–8 rooms +) and parish | Longitudinal multivariate survival analysis | Cox regressions | Age, sex, marital status | Meta analysis: Yes (occupation based social class) Apartment size and parish but not occupation-based social class was sign. |
| 34 | [51] | Spain (Andalusia, the Basque Country, Castile and Leon, Catalonia, Madrid, Navarre, and Valencia) | July 2009-Feb 2010 | Patients recruited from hospitals & primary health care clinics & emergency units during the peak of the influenza A 2009 pandemic in | 699 hospitalized and 699 non-hospitalized with Lab-confirmed cases A(H1N1) cases using (RT-PCR) | Individuals | Individuals | Cases filled in a questionaries' at the health centre or by phone to obtain covariate information | Hospitalized lab-confirmed A (H1N1) cases | Non-hospitalized (family physician visits at primary health care clinics and emergency units) cases of A (H1N1) infection | Education (Secondary or higher vs no formal education or primary education) and overcrowding (below the fifth percentile of the distribution of square metres available per person in the normal residence of all study participants) | Multicenter Matched case-control (according to age, date of hospitalisation in of the case (+/- 21 days) & province of the residence of the case) | Binomial logistic regression using Cox conditional logistic regressions | Sex, ethnicity, prior preventive information, prior pandemic vaccination, previous outpatient care or emergency care and unfavourable medical factors (smoking, morbid obesity (BMI >40), hypertension, lung disease, cardiovascular disease, kidney failure, diabetes, chronic liver disease, immunodeficiency, disabling neurological disease, malignancy, transplantation, cognitive dysfunction, seizure disorders and rheumatic diseases) | Meta analysis: Yes (education) Education decreases & Overcrowding increases outcome significantly |
| 35 | [14] | Global study covering 27 countries with high-quality vital registration data for the 1918–1920 pandemic | 1918–20 | Data for populations where vital registrations are believed to be more than 80% complete, supplemented with subnational data for US states & provinces of "pre-partition" India | 27 countries for 1918–1920, 24 US states with data available for the period, and nine Indian provinces | Countries and states | Aggregate | Human mortality database, B.R. Mitchels International Historical Statistics Series, subnational data from US states and provinces of prepartition India | Excess mortality by comparison of annual death rates during the pandemic to the average of annual death rates before and after the pandemic | Population at risk | Income (Per-head income in real international dollars (corrected for price changes) | Cross-sectional control-variable design | OLS with log of pandemic mortality and log income and absolute value of latitude | Latitude, to control for diurnal temperature fluctuation | Meta analysis: Yes Log per-head income in 1918 sign. |

(Continued)

**Table 1.** (Continued)

| * | Study authors and year | Study region | Pandemic period | Sample inclusion criteria | Sample size | Unit of outcomes | Data aggregation level | Data sources | Outcomes | Baseline outcomes | SES measures | Design | Statistical technique | Controls | Estimates used in meta analysis and is SES an independent predictor? |
|---|---|---|---|---|---|---|---|---|---|---|---|---|---|---|---|
| 36 | [52] | Canada (Ontario) | 13 April-20 July 2009 | Residents (children and adults) tested for A(H1N1) using RT-PCR | 240 cases and 112 controls among children (<18 years) and 173 cases and 229 controls among adults (>18 years) | Individuals | Individuals H1N1 status by individual education and several ecological SES variables | Clinic-based sample from Ontario, individuals presented to clinics for medical care + standardised telephone interviews | Lab-confirmed 2009 pandemic cases | RT-PCR negative H1N1 cases | Individual Education (high school or less and post-secondary school completion) Area measures: Material, social, total, low employment rate, low income. | Test-negative case-control study | Logistic regressions | age, gender, bmi, ethnicity, current smoker, underlying medical conditions, household density, children in household, receipt of 2008 seasonal vaccine, tested prior to 11th June 2009, healthcare provider, Toronto residence, immigrant category | Meta analysis: Yes (Total deprivation, one for adults and one for children). None of the SES variables were sign. in univariate models and were therefore not entered in the multivariate models. |
| 37 | [53] | Europa (30 EU/EFTA countries) | May 2009-May 2010 | Confirmed and notified fatal pandemic influenza A (H1N1) deaths in EU/EFTA region | 2896 fatal cases | Aggregate, Countries | Aggregate | ECDC and Eurostat | Lab-confirmed and notified deaths | Population at risk | GDP per capita | Cross-sectional control-variable design | Random effect Poisson regressions | greenhouse gas emissions, concertation of particular matter, latitude, hospital beds per 100,000 inhabitants, per capita government expenditure on health, unmet need for medical examination/ treatment, Gini coefficient, employment rate, proportion of population aged 65 +, old age dependency ratio, women per 100 men | Meta analysis: Yes GDP per capita was sign. in univariate model, but not in multivariate model. |
| 38 | [54] | Australia (Barwon statistical division in Southeastern Australia) | Sep 2009-May 2010 | Adult subjects in Geelong Osteoporosis Study, a group randomly selected from electoral rolls, were invited to participate in this sub-study to provide blood samples and complete a questionnaire. Sample of seropositive adults prior to the availability of a vaccine | 1184 individuals (129 seropositives and 1055 seronegatives) | Individuals | Individual seropositive status by ecological SES variables | Blood samples and self-report questionnaire | Haem agglutination inhibition test, seropositivity was defined as a titre > 1:40 | Seronegative persons | Australian Bureau of Statistics' Index of Relative Socioeconomic Advantage and Disadvantage (IRSAD) Area-level measure of education, occupation, income, unemployment and household structure (quintiles 1-5) | Cross-sectional control-variable design | Multivariate logistic regressions | age, bmi, obese, current smoker, healthcare worker, childcare worker/ teacher, employment status, highest level of education, lives alone, lives with children aged <12 years, chronic respiratory disease, pregnancy, chronic heart disease, diabetes | Meta analysis: Yes The SES variable was significant in multivariate models |

(*Continued*)

**Table 1.** (Continued)

| * | Study authors and year | Study region | Pandemic period | Sample inclusion criteria | Sample size | Unit of outcomes | Data aggregation level | Data sources | Outcomes | Baseline outcomes | SES measures | Design | Statistical technique | Controls | Estimates used in meta analysis and is SES an independent predictor? |
|---|---|---|---|---|---|---|---|---|---|---|---|---|---|---|---|
| 39 | [31] | England and Wales (62 of 82 counties) | Week ending 29 June 1918 to 10 May 1919 | Counties with SES info from 2000 which could be linked to counties in 1918 | Sample covers 333 units and 62 out of 82 counties | Individual deaths | Aggregate | Weekly influenza deaths & annualised rates/ 1000 population, collated by the Registrar General's Office in 1920 | Influenza mortality | Population at risk | The average of Ward Scores from the Indices of Deprivation 2000: District level Presentations for England It combines a number of indicators which cover a range of domains (Income, Employment, Health Deprivation and Disability, Education, Skills and Training Housing and Geographical Access to Services) into a single deprivation score for each area. | Cross-sectional control-variable design | non-parametric Spearman correlation coefficient | Pre-pandemic mortality, age, population size (persons/acre) | Meta analysis: No SES measure sign. in waves 1 and 3, but not wave 2 |
| 40 | [55] | USA (state of Massachusetts) | 26 April-30 Sep 2009 (before the vaccine became available) | Patients met the following inclusion criteria: 1) Patients were discharged from acute care hospital. 2) more diagnosis codes assigned 1 or corresponding to a grouping of ICD-9, 3) younger than 65 years | 4874 hospitalizations of which 526 admitted to ICU | Individuals | Individual hospitalizations, but area-level SES variables | Linked hospital discharge and American Community Survey and US Census data | Lab-confirmed H1N1 ICU stays | Hospitalized non-ICU patients | % of pop below poverty level 2006–2010 for zip code areas | Cross-sectional control-variable design | Logistic regressions | Racial/ethnic groups, gender, age, admission though EP/OP | Meta analysis: Yes Unexpectedly, those in less affluent SES groups had sign. lower risk of ICU stay than the most affluent SES group |
| 41 | [56] | USA (341 US counties in 14 states) | July 2009-June 2010 | Only states with consistent reporting and updating of H1N1 statistics, that is reporting standards met by the CDC | Sample size not given. | Aggregate,341 counties | Aggregate | County-level H1N1 deaths are from CDC and SES variables from US census and CDC 11% of US counties covered, SES measures are representative to similar characteristics to USA as a whole | H1N1 deaths according to CDC | Population at risk | Per capita personal income; median household income; educational attainment (persons aged >/ = to 25 years), percent high school graduate or higher, educational attainment (persons aged >/ = 25 years), percent bachelor's degree or higher; people of all ages in poverty (%) | Univariate and cross-sectional design | Correlations | No controls | Meta analysis: Yes In univariate models poverty positively predicted mortality while income and education variables negatively predicted mortality. Multivariate modelling was not carried out. |

*(Continued)*

Table 1. (Continued)

| * | Study authors and year | Study region | Pandemic period | Sample inclusion criteria | Sample size | Unit of outcomes | Data aggregation level | Data sources | Outcomes | Baseline outcomes | SES measures | Design | Statistical technique | Controls | Estimates used in meta analysis and is SES an independent predictor? |
|---|---|---|---|---|---|---|---|---|---|---|---|---|---|---|---|
| 42 | [57] | Spain (Andalusia, the Basque Country, Castile and Leon, Catalonia, Madrid, Navarre, and Valencia) | July 2009-Feb 2011 | Cases and controls were aged > 18 years and picked from 36 hospitals and 22 primary-care centres | 715 primary care H1N1 cases, 715 other diseases than ILI primary centre controls, and 406 hospitalized H1N1 cases | Individuals | Individuals | Hospital and primary care data | Lab-confirmed H1N1 cases and hospitalizations (RT-PCR) | Infection model: Controls were primary care patients with other disease than ILI. Hospitalization model: cases were primary care centre H1N1 cases | Occupational based social class (Manual vs. non-manual workers) | Matched case-control study | Logistic regressions | In model for infection: age, pregnancy, diabetes and influenza vaccination. In hospitalization model: age, pregnancy, COPD, cardiovascular disease, diabetes, and influenza vaccination | Meta analysis: Yes SES variable sign. in multivariate models for both infection and hospitalization risks |
| 44 | [17] | England | 1 June 2009-18 April 2010 | All deaths reported due to pandemic flu | 349 out of 365 deaths (95.6%) in England | Individual deaths | Aggregate: Individuals aggregated up to five approximately equal population groups to create area deprivation quintiles | National Health Service; basic set of demographic information | Pandemic deaths, no info whether these were lab-confirmed or not, but they were probably lab-confirmed | Population at risk | Index of Multiple Deprivation of an area and postcodes (pooled measure based on income, education, housing, health and crime) (1–5, where 5 is least deprived and 1 most deprived) | Cross-sectional table analysis | Direct age-sex standardization of mortality rates using mid-point 2009 pop estimates for England | Age, sex, and Urban and rural areas | Meta analysis: Yes SES variable significant with and without urban-rural interactions |
| 45 | [58] | Global: 20 countries covering 35% of the world population | 2009 pandemic mortality | Weekly virology and underlying cause-of-death mortality time series for 2005–2009 | 123,000–203,000 deaths in the last 9 months of 2009 | Aggregate | Aggregate | Weekly virology data from the WHO FluNet and national mortality time series | Excess mortality associated with the 2009 pandemic | Population at risk | Gross national income (GNI) per capita (US dollars) | Univariate cross-sectional time-series analysis | Multivariate OLS regressions | - | Meta analysis: No. Coefficients not given in the paper or in online appendix. Estimates between Gross national income and mortality was ns. |
| 48 | [28] | New Zealand | 27 Aug 1918-March 1919 | Male soldiers (New Zealand Expeditionary Forces (NZEF) in both hemispheres in 1918–1919 pandemic period) | 930 deaths, taken from 1000 randomly selected records | Individuals | Individuals | Death certificates | Influenza, pneumonia, and bronchitis deaths | NZEF population at risk | Pre-enlistment occupational based social class (1–3 (most privileged), 4–6 and 7–9 (least privileged) | Univariate cross-sectional design | Univariate Rate ratios | No controls | Meta analysis: Yes SES measure not significant |
| 49 | [29] | New Zealand | 20 July-13 Oct 1918 | Male navy soldiers (military personnel in HM New Zealand Transport troop ship Tahiti) | 77 deaths, 1117 military personnel plus 100 crew (total pop at risk 1217) | Individuals | Individuals | Death certificates | Influenza and pneumonia deaths | Population at risk at HM New Zealand Transport troop ship Tahiti | Occupation-based social class (1–6 and 7–9 (1 is company manager and 9 is labourer) | Cross-sectional control-variable design | Multivariate logistic regression | age, military rank, rurality score, military unit | Meta analysis: YES SES measure not significant |

(Continued)

**Table 1.** (Continued)

| * | Study authors and year | Study region | Pandemic period | Sample inclusion criteria | Sample size | Unit of outcomes | Data aggregation level | Data sources | Outcomes | Baseline outcomes | SES measures | Design | Statistical technique | Controls | Estimates used in meta analysis and is SES an independent predictor? |
|---|---|---|---|---|---|---|---|---|---|---|---|---|---|---|---|
| 50 | [30] | USA (New London, Connecticut, Baltimore, Maryland, Augusta, Georgia, Macon, Georgia., Des Moines, Iowa, Louisville, Kentucky, Little Rock, Arkansas, San Antonio, Texas, San Francisco, California | 1 Sep-Dec 1918 | Nine urban localities with a population of at least 25,000, randomly selected, only white populations | 94,678 individuals, 26,824 morbidity cases (influenza, pneumonia and "doubtful" cases), X deaths | Individuals | Aggregate | Survey data (e.g. Baltimore: sample 33,776 (5.68% of pop) | Self-reported pandemic morbidity, mortality and case fatality rates ( | Morbidity: Population at risk in canvassed areas and lethality: mortality among the sick | Economic status (Very poor; poor; moderate; well-to-do (based on the enumerators impression) | Cross-sectional control-variable design | Cross-tables and direct standardization techniques to control for age-differences etc. | age, sex, size of household | Meta analysis: Yes SES measure sign. related with both outcomes. |
| 51 | [16] | USA (New Haven County, Connecticut) | 2009–10 | Hospitalized, laboratory confirmed influenza among adults 18 years and older | 213 hospitalizations | Individuals | Individual lab-confirmed hospitalizations but neighbourhood level SES measures (185 Census tracts) | Surveillance data (Connecticut Emerging Infections Program's influenza-associated hospitalisation surveillance system) + chart reviews & interviews with healthcare providers & with patients or their proxies. Census tract level data obtained from the US Census Bureau's 2006–2010 American Community Survey (ACS) | H1N1 lab-based hospitalizations | Population at risk in New Haven | Below federal poverty, no high school diploma, median income | Cross-sectional design | Age-adjusted incidence of influenza-associated hospitalizations among adults by neighbourhood SES characteristics. | Age. | Meta analysis: Yes All 3 SES measures are sign. and display a clear social gradient |
| 52 | [59] | USA (state of New Mexico) | 14 Sep 2009–13 Jan 2010 | Hospitalized, positive influenza hospitalization, Mechanical ventilation and death among the hospitalized | 926 lab-confirmed H1N1 hosp. Patients, 106 mechanically ventilated and 35 deaths | Individuals | Individuals outcomes, but 33 counties divided into 4 quartiles by median household income | New Mexico Department of Health statewide surveillance of hospitalizations and deaths. Estimates from the US Census Bureau's Small Area Income and Poverty Estimates programme. | H1N1 related hospitalisations, mechanical ventilation and death | Comparison group for hospitalizations: general statewide population; Comparison group for mechanical ventilation and death among those hospitalized were the hospitalized | Household Income (County median household annual income quartile) | Cross-sectional control-variable design | Poisson and logistic regressions | Hospitalization model: age, gender, and race/ethnicity. Mechanical ventilation model: age, gender, and race/ethnicity, obesity, high risk conditions, neuraminidase treatment, time from illness onset to seeking medical care. Mortality risk model: ns in unadjusted model, therefore no multivariate model | Meta analysis: Yes SES measure sign. in model for hospitalization risk but not in models for mechanical ventilation and death |

(*Continued*)

Table 1. (Continued)

| Study authors and year | Study region | Pandemic period | Sample inclusion criteria | Sample size | Unit of outcomes | Data aggregation level | Data sources | Outcomes | Baseline outcomes | SES measures | Design | Statistical technique | Controls | Estimates used in meta analysis and is SES an independent predictor? |
|---|---|---|---|---|---|---|---|---|---|---|---|---|---|---|
| 53 [60] | Canada (Winnipeg, Manitoba) | Oct- Dec 2009 | Adults presenting to three inner city community clinics were recruited as study participants using convenience sampling. | 458 study participants (174 participantsOct-12 Nov, before the vaccine was available), 206 cases 13 Nov-Dec, which did not get take the vaccine; 78 participants enrolled on or after Nov 13 which did get the vaccine are not included in our meta-analysis) | Individuals | Individuals | Serological testing and questionnaire data | Seropositive cases | convenience sample population at risk | Education (High school or not) and annual household income | Univariate & cross-sectional analysis | Prevalence estimates with exact binomial 95% CI using Clopper Pearson intervals | no controls | Meta analysis: Yes The two SES measures ns. for both periods. |
| 55 [61] | Australia (Northern Territory) | 2009 (June-August) | Antibody titers were determined by hemagglutination inhibition against reference virus A/California/7/2009 on serum samples collected opportunistically from outpatients | 1689 serologic specimen post pandemic (cases 3–30 September 2009) and 445 serological specimen prepandemic (controls January 10 to May 29, 2009) | Individuals | Individual seropositive status but SES measure is aggregate | Serological data, specimens from pathology lab, and computer matching of data to indigenous status and SEIFA measures | lab-confirmed seropositivesand attack rates (difference between post and pre-pandemic immunity) | serological specimen prepandemic (controls January 10 to May 29, 2009) | 2006 Statistical Local area (SLA) was linked to Australian Bureau of Statistics' Socio-Economic Indexes for Ara (SEIFA). SEIFA measures (quintiles) use information from census data relating to material and social resources and ability to participate in society to obtain a broad level of relative socioeconomic status for each SLA | Case-control design | Logistic regressions | age, gender, aboriginal and Torres strait islanders, region | Meta analysis: Yes SES measure ns. |
| 57 [62] | Canada (province of Manitoba) | 2 April-5 Sep 2009 | Confirmed H1N1 cases for whom the final location of treatment was known | 795,569 community cases, 181 hospitalized but not ICU, 45 admitted to ICU | Individuals | Individual H1N1 case status, but area income quintiles | Lab–confirmed H1N1 data, hospital data and data collection–form completion via interviews | lab-confirmed community cases, hospitalizations and ICU admissions | Two control groups. Community cases (vs. hospitalizations) and hospitalized, non ICU (vs. ICU). | Income based on postal codes (Top three quintiles vs the bottom two quintiles) | Cumulative case-control design | Logistic regressions | Age, gender, pregnancy, ethnicity, any comorbidity, Interval from symptom onset to antiviral treatment, rural vs urban | Meta analysis: Yes SES measure ns. in models for both hospitalizations and ICU admissions |

*(Continued)*

**Table 1.** (Continued)

| * | Study authors and year | Study region | Pandemic period | Sample inclusion criteria | Sample size | Unit of outcomes | Data aggregation level | Data sources | Outcomes | Baseline outcomes | SES measures | Design | Statistical technique | Controls | Estimates used in meta analysis and is SES an independent predictor? |
|---|---|---|---|---|---|---|---|---|---|---|---|---|---|---|---|
| 58 | [63] | China (Beijing) | 1 Aug–30 Sep 2009 | Households of hospital healthcare workers. Case households were: (1) has an index patient of H1N1. (2) index case was quarantined in household from onset of diagnosis to 7 days after onset of illness; (3) secondary case had potential contact with index patient; (4) symptoms onset of secondary case occurred within 7 days since last known contact with index case during infectious period of index case; (5) RT-PCT confirmation date of secondary case occurred within 7 days since last known contact with index case during infectious period of index case; (6) none of the household members previously received a vaccine against pandemic H1N1 2009 influenza | 54 case households (HH with a self-quarantined index patient and a secondary case), 108 control households (HH with a self-quarantined index patient and a close contact) | Individuals | Households | Household transmission data | Lab-confirmed secondary cases (RT-PCT) | Households with a self-quarantined index patient and a close contact | Education (High school and higher vs middle school and lower) | 1:2 matched case-control design | Conditional logistic regression | Sharing room with index case-patient; Ventilating room every day; and Frequency of hand washing | Meta analysis: Yes SES measure significant |
| 59 | [64] | England | 27–30 April 2009 | Lab-confirmed AH1N1 pandemic flu deaths | 337 of 389 lab-confirmed fatalities (86.6%) | Individuals | Individual lab-confirmed deaths, but SES is measured for 32378 super output areas (LSOA) | National Health Service | lab-confirmed deaths | Population at risk | Index of Multiple Deprivation of an area and postcodes (IMD 2007). It includes seven dimensions: income, employment, health deprivation and disability, skills and training, barriers to housing and services, crime and disorder, living environment | Cross-sectional control-variable design | Poisson regressions | Age, gender, rural vs urban | Meta analysis: Yes SES measure significant. |

* These numbers correspond to the 59 studies from which we extracted data. In the data extraction phase, we removed an additional 15 studies The final number of studies included in the narrative synthesis was therefore the 44 listed in this table, also see documentation in S2 File.

unduly influence the result, the Bayesian model is also estimated without study-level covariates to allow for comparison with the standard random effect model.

## Results

### Narrative review

**Flow of included studies.** Our database search identified 8,411 records. After leaving out duplicates, 4,203 studies were imported for screening. After removing another 75 duplicates, we screened the titles/abstracts of 4,128 records. Of these, 3952 studies were irrelevant, and 176 full text studies were then assessed for eligibility. In this phase, 117 studies were excluded, leaving us with 59 studies from which to extract data. In the data extraction phase, we removed an additional 15 studies. The final number of studies included in the narrative synthesis was therefore 44 (see PRISMA Flow Chart in S1 File).

**Study characteristics.** The review identified a total of 44 studies, 9 studies of "Spanish flu of 1918–20" [12–14, 26–31] and 35 of the "Swine flu of 2009–2010" [15, 16, 32–64] (Table 1). We identified no studies of the Russian flu of 1889–90, the Asian flu of 1957–58 or Hong-Kong flu of 1968–70. Most of the studies used data from North America, including 11 for USA [16, 27, 30, 33, 40, 41, 48, 50, 55, 56, 59] and 6 for Canada [38, 45, 49, 52, 60, 62]; Europe, including 6 for England [15, 26, 31, 32, 44, 64], 4 for Spain [39, 46, 51, 57], 2 for Norway [12, 13], and 1 for 30 EU/EFTA countries [53]; 4 for Australia [42, 43, 54, 61] and 3 for New Zealand [28, 29, 34]. While a few studies used data from Central America/South America including 1 for Mexico [37] and 1 for Brazil [47], and Asia, including 1 for Iran [35] and 1 for China [63], we identified no studies using data from Africa. Finally, 3 studies had a global approach studying several countries [14, 36, 58].

The sample inclusion criteria varied greatly from study to study. Two of the 44 studies studied military populations, one of these studied mortality in randomly selected records [28], the other studied mortality on one transport troop ship [29]. Of the 42 studies using civilian study populations, some studied particular patient populations/cohorts [46, 54, 61, 63], general patients at various hospitals and health centres [16, 32, 33, 35, 39, 40, 47–49, 51, 52, 55, 57, 59, 60, 62], students at schools or students including their families [41, 45], or general populations living in various cities, states, counties or (several) countries [12–15, 26, 30, 31, 34, 36–38, 42–44, 50, 53, 56, 58, 64].

The sample size in each study varied substantially and is reported in Table 1 whenever information was available for the pandemic events (for cases and controls) and the population at risk.

The unit of the outcome variables was either individual in 36 studies [13, 15, 16, 27–40, 42–49, 51, 52, 54, 55, 57, 59–64] or aggregate in 8 studies [12, 14, 26, 41, 50, 53, 56, 58]. Some of the studies with individual-level outcome data nevertheless preformed analysis at aggregated levels. In 12 studies the data aggregation level was aggregate [12, 14, 15, 26, 30, 31, 36, 41, 50, 53, 56, 58]. 15 studies had individual-level outcome variables and control variables, but used area-level (and individual-level) SES variables [16, 27, 32, 33, 42–44, 49, 52, 54, 55, 59, 61, 62, 64]. Studies using only ecological SES variables thus picked up a combination of individual-level and area-level SES effects on the outcome variables. Finally, in 17 of the studies, outcomes, explanatory variables and controls were all measured for individuals and the data aggregation level was thus the individual level [13, 28, 29, 34, 35, 37–40, 45–48, 51, 57, 60, 63].

There were generally three types of data source used in the 44 studies included in the narrative synthesis: 1) 28 studies used active surveillance of events coupled with SES and covariate data via questionnaires, face-to-face or telephone interviews or censuses [16, 32–44, 46–52, 54, 57, 59–63]; 2) 14 studies used national vital registration systems on events coupled with SES

and covariate data via censuses [12–15, 26–29, 31, 53, 55, 56, 58, 64]; 3) 2 studies used telephone survey or data collected via door-to-door survey to collect both event and population at risk data [30, 45].

The 3 broad categories of outcomes were studied (see details in Table 1): 1) people seeing doctors due to symptoms of influenza like illness (ILI)/influenza transmission(R0)/lab-confirmed influenza infection (using PCR tests)/immunity towards influenza (using blood serum samples to look for antibodies) [26, 27, 30, 32, 34, 35, 41–45, 52, 54, 57, 60, 61, 63]; 2) lab-confirmed influenza hospitalizations/ICU treatment/mechanical ventilation [16, 33, 38, 39, 46–51, 55, 57, 59, 62]; 3) lab-confirmed pandemic deaths/Influenza-Pneumonia (PI) deaths/excess deaths associated with pandemic influenza [12–15, 26–31, 36, 37, 40, 53, 56, 58, 59, 64].

The choice of baseline outcomes (or controls in case-control studies) partly depended on the outcomes studied, and included: 1) General population at risk [12–16, 26–33, 36, 50, 53, 56, 58–60, 64]; 2) General population at risk without H1N1 Infection or ILI [41–45]; 3) Patients with ILI, persons in quarantine for a suspected case and a close H1N1 contact or patients with ILI testing negative for influenza A H1N1 infection [30, 35, 52, 63]; 4) pre-pandemic immunity [34, 61]; 5) seasonal influenza A deaths [37]; 6) Non-hospitalized H1N1 positive patients or hospitalized H1N1 positive non-severe (not ICU or death) [38, 39, 55, 59, 62]; 7) Outpatients with H1N1 infection [40, 46–49, 51, 57]; 8) Seronegative for H1N1 [54]; 9) Patients with other diseases than ILI [57].

The studies that used individual-level SES measures used one or several of the following; (household) income [40, 48, 60], economic status [30], education [35, 37–40, 46–49, 51, 52, 60, 63], occupation-based social class [13, 28, 29, 57], size of apartments, poor housing or crowding measures [13, 26, 34, 40, 45, 49, 51], and having health insurance [40]. Some used both individual-level and area-level measures of SES. The SES measures used at the area-level were often (but not always) indexes of economic, social and housing deprivation/development [12, 14–16, 27, 31–33, 36, 41–44, 48–50, 52–56, 58, 59, 61, 62, 64].

The 44 studies included in the review used study designs that fall into four categories: 1) Systematic review and meta-analysis [36]; 2) Cross sectional univariate or control-variable design [12, 14–16, 26–34, 42–45, 50, 53–56, 59, 60, 64]; 2) Case-control design [35, 37–41, 46–49, 51, 52, 57, 61–63]; 3) Longitudinal survival analysis [13]; 4) Time-series analysis [58].

The identified studies were descriptive or explanatory. The descriptive studies used statistical techniques to calculate pandemic disease burden estimates and univariate correlations between the outcomes and various variables as well as demographic standardization techniques to control for age and sex [15, 16, 28, 30–33, 44, 56, 60]. The explanatory multivariate studies used modelling techniques such as OLS [12, 14, 50, 58], generalized linear mixed models [45], logistic regressions [26, 29, 34, 35, 38–41, 46–49, 52, 54, 55, 57, 59, 61–63], propensity score logistic regressions [37], Poisson regressions [27, 53, 59, 64], Cox regressions [13, 51], random effect meta-regressions [36], and various types of Bayesian models [42, 43].

**Study results.** The results in the 9 identified studies on the 1918 influenza and SES were mixed [12–14, 26–31]. After various controls were made, 6 studies found a significantly and expected *higher* mortality for *lower* SES groups [14] or higher mortality/transmission rates, but not a clear social gradient for all SES measures [12, 13, 27]; a significant *higher* mortality for *lower* SES groups, but only for 2 out of 3 pandemic waves [31]; or a significantly *higher* morbidity and mortality for *lower* SES groups [30], while 3 studies found no association between SES and mortality [28, 29] or mortality and transmission rates [26]. However, none of the 6 studies documenting significant associations with a higher pandemic disease burden for lower SES groups included data to control for medical risk factors. Hence, some or all of the identified associations between SES and the pandemic outcomes in the 6 above mentioned studies

could potentially have been "explained away" by controlling for having latent tuberculosis [65] or other known comorbidities [66].

Fourteen of the 35 identified studies on the 2009 pandemic had data to adjust for both medical and social risk factors [34, 35, 38, 40, 45–48, 51, 52, 54, 57, 59, 62]. After adjusting for medical risk factors, 7 of these studies documented independent and *expected* impact of SES (*higher* risks for *lower* SES) on either infection/immunity [34, 54], hospitalization [46–48, 51] or both of these outcomes [57]; 1 study found both expected significant associations with SES (higher risk of hospitalization) and non-significant (ICU and death) impact of SES after medical risk factor were controlled for [59]; 5 studies found non-significant effects of SES on ILI/ infection/immunity [45, 52], hospitalization/ICU [38, 62] and mortality [40]; and finally, 1 study found a significant but *unexpected* impact of SES on infection, that is *higher* infection rates for those with *high*er vs. lower education [35]. Although the findings in these 14 studies investigating both social and medical vulnerabilities were somewhat mixed, they show that medical risk factors are not simply 100% correlated with socioeconomic factors, and in 8 of these 14 studies social factors explained variation in the variation in the pandemic outcomes beyond that captured by medical factors.

21 of the 35 identified studies on the role of SES in the 2009 pandemic outcomes *did not* control for medical risk factors but found the following. First, 12 studies found significantly *higher* risks for the *lowest* socioeconomic status group, of which 5 studied ILI/infection/immunity [32, 43, 44, 63]; 4 investigated hospitalizations [16, 33, 39, 49]; and 4 studied mortality [15, 36, 56, 64]. Second, 7 studies found non-significant associations with SES, of which 2 studied ILI/infection/ immunity [42, 60]; 2 studied hospitalizations [50, 62] or ICU treatment [62], and 3 studied mortality [37, 53, 58]. Finally, 2 studies unexpectedly found respectively a higher risk of a lab-confirmed case [41] or ICU treatment [55] in the *highest* SES groups. It is clear though, that most of the studies on SES and 2009 pandemic not controlling for medical at risk factors [13 of 21], showed that lower SES groups have the highest risks of the three considered pandemic outcomes.

## Quantitative synthesis

The quantitative synthesis includes 46 estimates drawn from 35 of the 44 studies included in the narrative synthesis [13–16, 27–30, 32–35, 37–41, 45–49, 51–57, 59–64], and a standard random effects analysis of all estimates pooled found a pooled effect mean odds ratio of 1.4 (95% CI: 1.2–1.7), comparing the low to the high SES groups. The pooled estimate was statistically significant at the 0.1 percent level, which means that we would have been highly unlikely to see an estimate of this or larger absolute magnitude if the true mean of the effect distribution was zero. As seen in the forest plot, the individual study estimates differed in both precision and location, with more variation in less precise estimates as we would expect (Fig 1). To test for the presence of publication bias, we used Egger's test and Begg's test as implemented in the regtest and ranktest commands of the Metafor R-package [23].As imprecise estimates have to be larger to be statistically significant, publication bias will tend to show up as a systematic relationship between point estimates and their standard errors. Neither a rank-rank correlation test (Begg's test, p-value 0.68) nor a regression test (Egger's test, p-value 0.81) indicated any such relationship.

In the random effect analysis we found strong evidence of effect heterogeneity across studies, with an estimated 92% of the total variation across studies reflecting effect differences rather than sampling variation. The estimated standard deviation of the effect distribution (tau) has a point estimate of 0.45 on the log scale. If the underlying effects at the study level are normally distributed around their expectation, this tau is the estimated standard deviation of study effects. The estimates would then imply that there is a 50% chance that the true

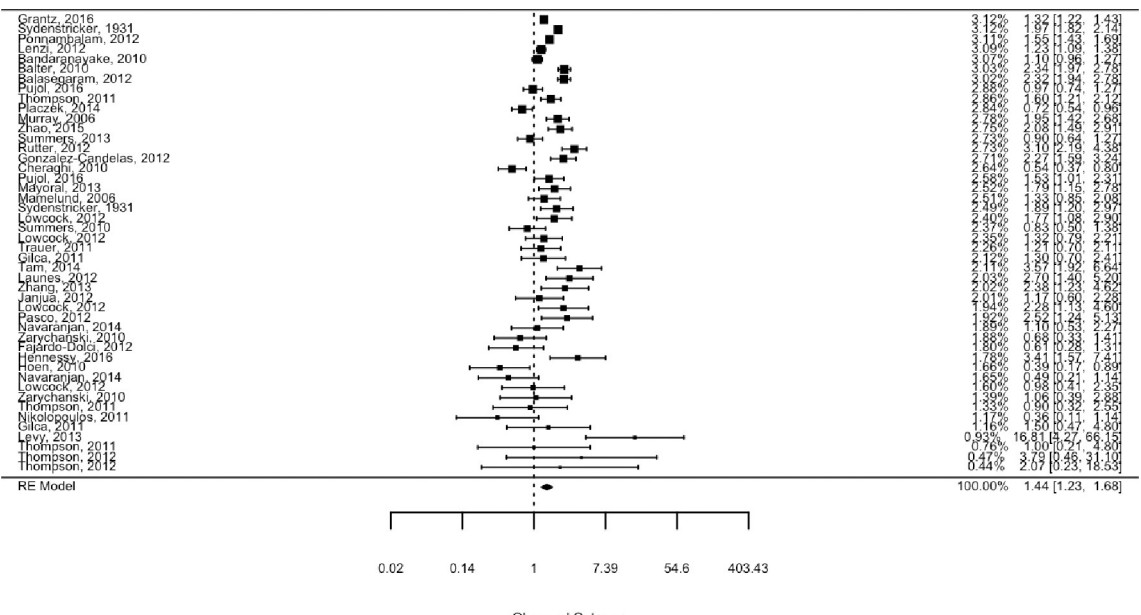

**Fig 1. Forest plot.** The plot shows the included estimates sorted by precision, along with their weights in the pooled effect estimate.

parameter value of a randomly selected study lies in the range of 1.1–1.9. The Cochran's Q test strongly rejects a test of zero heterogeneity (p < 0.0001), confirming the choice of a random effects over a fixed effect model. Our subsample analyses indicate similar results in studies using individual level and aggregate SES indicators, case control and relative risk outcome measures, and studying the 1918 and 2009 pandemic period (Fig 2 and Table 2).

Subsamples were also defined by specific *combinations* of case and control outcomes (Fig 3). These suggest that studies examining the risk of flu outcomes relative to a general healthy population (here defined as a control sample not selected on indicators of illness) tend to indicate a clear and substantial increased risk for lower SES groups. Studies comparing hospitalized to those infected also point to increased risks for lower SES groups. Studies assessing the risk of severe cases (e.g., treatment in ICU or death) *conditional* on hospitalization are fewer, but seem to report no clear SES associations in any direction. Finally, studies using "other" control samples (e.g., patients with flu symptoms who did not have flu, people with non-pandemic flu during a pandemic period, patients accessing or being treated by health care systems for other reasons) tend to find no (or reversed) associations with SES indicators.

As all of these comparisons were based on different splits of the same study sample, they can be viewed as a series of univariate analyses. To assess the joint contribution of these study level features, and to include country/region indicators, we estimated two Bayesian models: One, without study level covariates, was closely analogous to the above meta-analysis, and was included to ensure that results from the two approaches were similar and comparable. This Bayesian model finds a pooled effect mean of 1.4 with a 95% credibility interval from 1.2–1.7, which is identical to the above estimate of 1.4 (95% CI: 1.2–1.7). The estimated standard deviation of the underlying study parameters, analogous to the parameter tau in the earlier analysis, is estimated at 0.46 (0.3–0.6), the same as the above estimated tau of 0.45 (See S2 File for model code and discussion of prior choices). The second Bayesian model includes all study level indicators (level of SES indicator, RR/OR indicator, period, case and control outcomes, and country/region), as well as an indicator for each unique *combination* of case and control outcome (as in Fig 3). Jointly, this reduces the estimated unexplained heterogeneity (tau) substantially,

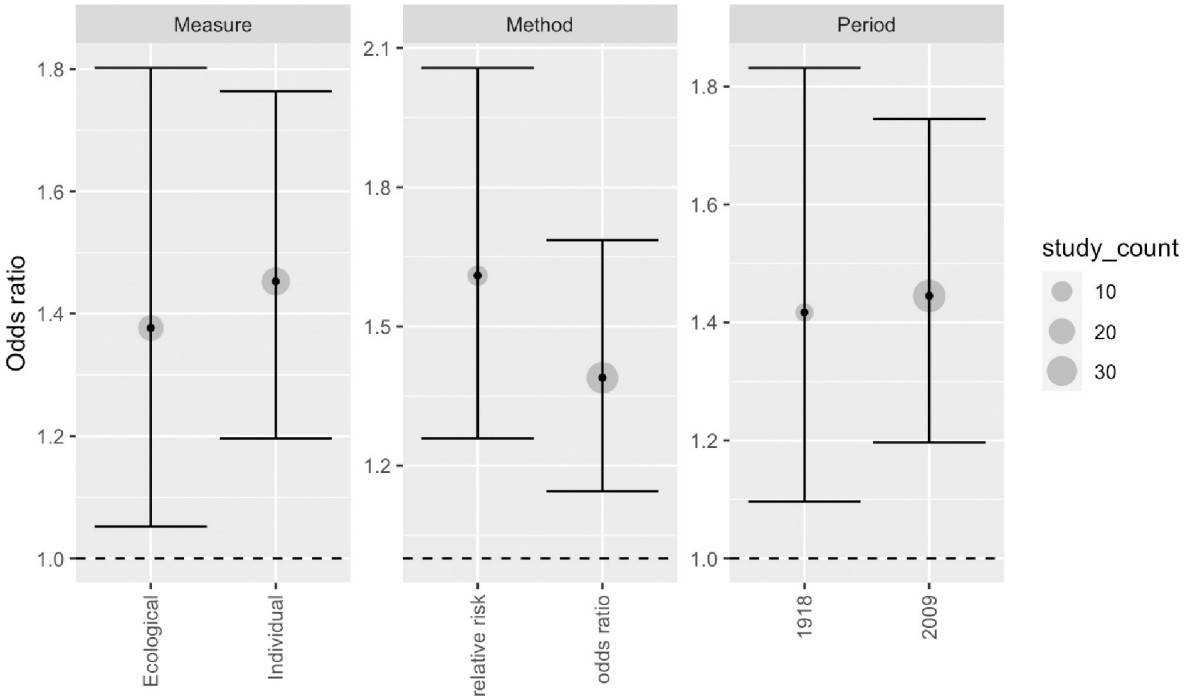

**Fig 2. Subsample analyses.** The plot shows point estimates and 95% confidence intervals for different subsamples of studies, with a grey circle indicating the number of studies in each subsample.

with the average value estimated dropping from 0.46 to 0.34 (See S2 File for model code and discussion of prior choices).

As shown in Figs 4 and 5, the Bayesian analysis finds similar results as the earlier subsample analyses, indicating that the patterns for the control and treatment outcome combinations are not "explained away" in an analysis when simultaneously accounting for other study level characteristics.

## Discussion

Research on Covid-19 has shown that the disease burden differs by SES, race and ethnicity [2–5]. This is consistent with the results we report here from the first systematic literature review

**Table 2. Subsample analyses.**

| Distinction | Type | Number of estimates | Pooled RE effect | 95% CI lower bound | 95% CI upper bound | Tau |
|---|---|---|---|---|---|---|
| Measure | Ecological | 20 | 1.38 | 1.05 | 1.80 | 0.53 |
| | Individual | 26 | 1.45 | 1.20 | 1.76 | 0.41 |
| Period | 1918 | 7 | 1.42 | 1.10 | 1.83 | 0.30 |
| | 2009 | 39 | 1.44 | 1.20 | 1.76 | 0.50 |
| Method | Relative Risk | 10 | 1.61 | 1.26 | 2.06 | 0.35 |
| | Odds Ratio | 36 | 1.39 | 1.14 | 1.69 | 0.49 |

The plot shows point estimates and 95% confidence intervals for different subsamples of studies, with a grey circle indicating the number of studies in each subsample.

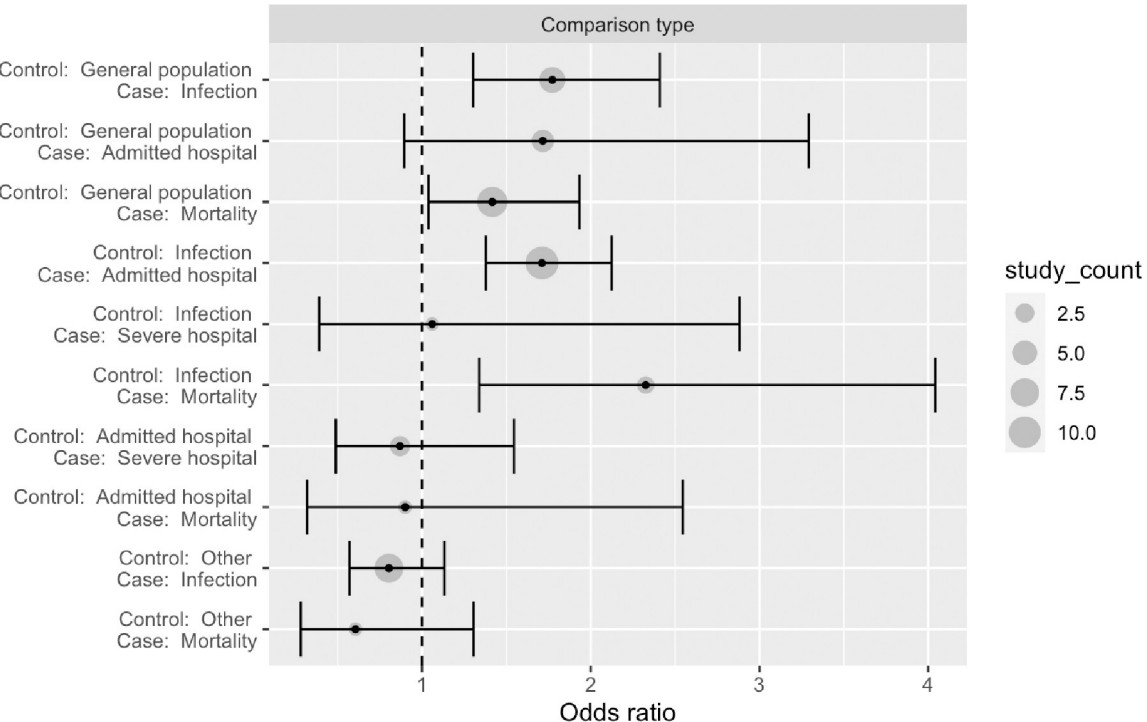

**Fig 3. Subsample analyses.** The plot shows point estimates and 95% confidence intervals for different subsamples of studies, with a grey circle indicating the number of studies in each subsample.

on the associations between SES and disease outcomes in the last 5 influenza pandemics. We identified nine studies of the "Spanish flu of 1918–20" and 35 of the "Swine flu of 2009–2010", but no studies of the "Russian flu" pandemic of 1889–90, the "Asian flu" of 1957–58 or the "Hong-Kong flu" of 1968–70. Most of the studies included for the 1918 and 2009 influenza pandemics used data from western high-income countries. Out of 51 estimates from 35 studies, the overall pooled mean pandemic outcome odds ratio was 1.44 (95% CI: 1.23–1.68) comparing the lowest to the highest SES groups. As expected, the random effect model estimated substantial effect heterogeneity across studies, which means the pooled effect mean should not be taken as valid at the single-study level. Based on the model, we would expect about 50% of the underlying effects to be in the 1.1 to 1.9 range. There was no evidence suggesting differences by pandemic period (1918 or 2009), the level of SES measure (individual or ecological), or type of method (odds ratio or relative risk). Finally, studies using healthy controls tended to find low SES associated with worse influenza outcome, and studies using infected controls found low SES associated with more severe influenza outcomes. Studies comparing severe outcomes (ICU or death) to hospital admissions were few but indicated no clear association. Studies with more unusual comparisons (e.g., pandemic vs seasonal influenza, seasonal influenza vs other patient groups) reported no or negative associations. These patterns were similar in a multivariate Bayesian model accounting for all study level indicators simultaneously. The Bayesian model also included indicators for study region/country. Relative to the "across all country/regions" average, studies from Australia, UK and to a lesser extent the USA tended to report stronger associations in our sample, while New Zealand tended to report weaker associations. These country-level results should be viewed as exploratory: two of the three studies from New Zealand [28, 29], for instance, were studies of how pandemic influenza outcomes varied across pre-service occupational status amongst military personnel during the 1918

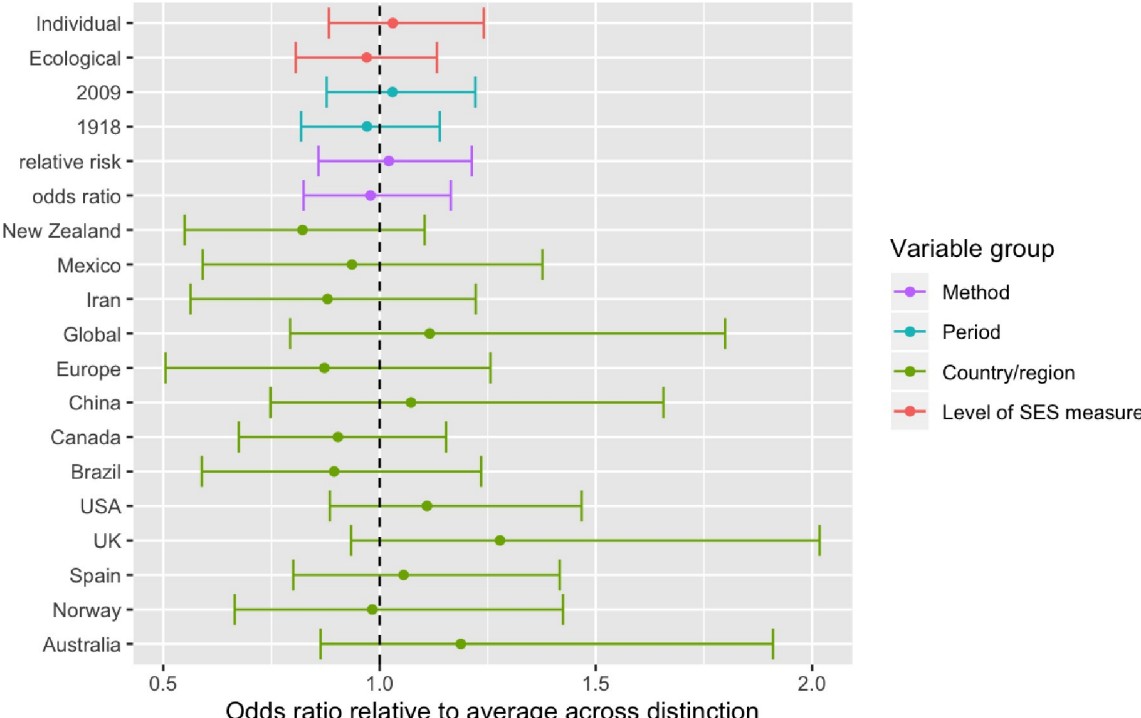

**Fig 4. Differences across study level covariates.** The plot shows average estimates and 95% credibility intervals for different study level covariates. The parameters are constrained to sum to zero within each category (e.g., for each draw from the posterior distribution, the sum of country parameters will sum to zero, as will the sum of the period parameters, etc.) See S2 File for model details.

pandemic, which are unlikely to speak broadly to such associations in New Zealand more generally.

Our results provide strong evidence that social risk factors matter for pandemic influenza outcomes in addition to medical risk factors. We also documented that in the 2009 pandemic, social risk factors independently explained variation in disease outcomes even when medical risk factors were controlled for [34, 46–48, 51, 54, 57, 59]. This is similar to the finding of a study of COVID-19 hospital deaths demonstrating that medical risk factors did little to explain the higher risks of the deprived and of immigrants in the UK [3]. Although we did not find support for our hypothesis that social disparities would be larger for more severe (e.g. ICU and death) than less severe outcomes (e.g. infection or hospitalization not requiring ICU), the similarity of results for the 1918 and 2009 pandemics show the persistence of individual- and ecological-level social risk factors, although the specific mechanisms and types of social vulnerabilities leading to social disparities in pandemic outcomes may differ between 1918 and 2009, or in 2020 during the COVID-19 pandemic. Results from this review on pandemic influenza and results from studies on the role of social and ethnic vulnerability in COVID-19 disease outcomes [2–5], support recent calls for the inclusion of social and ethnic vulnerabilities in addition to medical at risk factors in pandemic preparedness plans [18]. Examples are the prioritization of vaccines for medically vulnerable people living in socially vulnerable areas (urban slums or hard-to-reach groups in rural and remote areas), or SES groups with undiscovered medical vulnerabilities, and others who are at significantly higher risk of severe disease or death (various indigenous, ethnic, or racial groups, people living in extreme poverty, homeless and those living in informal settlements or; low-income migrant workers; refugees,

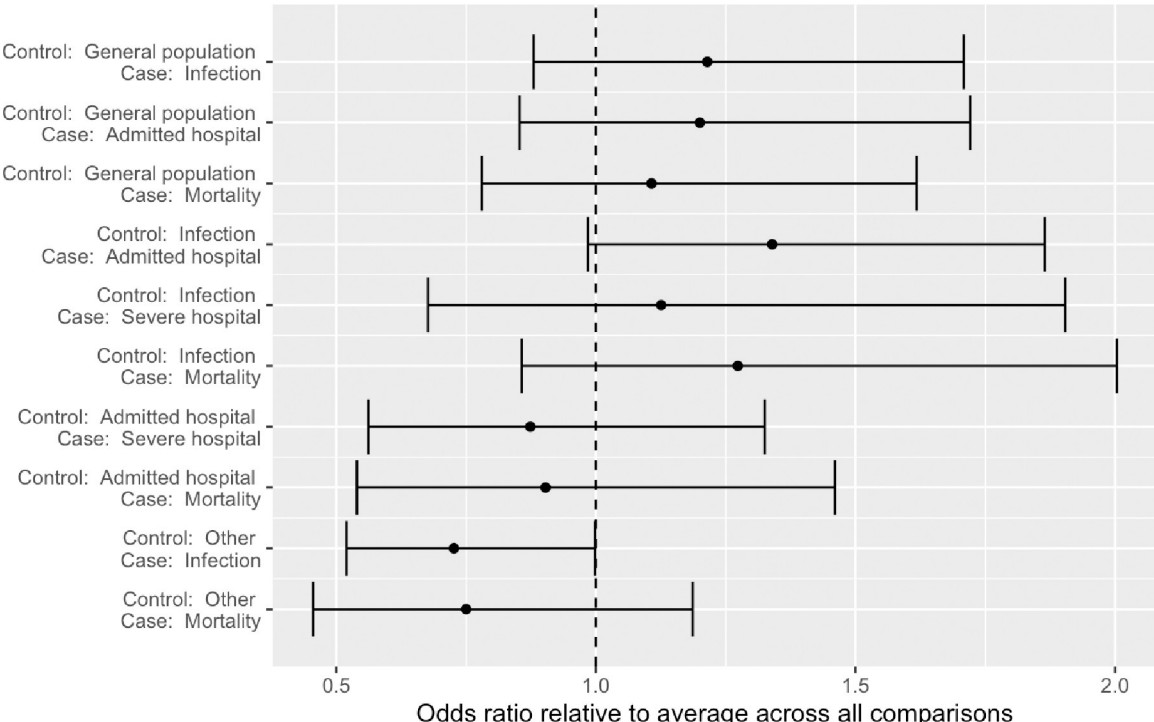

**Fig 5. Differences across case and control outcome combinations.** The plot shows average estimates and 95% credibility intervals for all combinations of case and control outcomes observed in the data. See S2 File for model details.

internally displaced persons, asylum seekers, populations in conflict settings or those affected by humanitarian emergencies, vulnerable migrants in irregular situations and nomadic populations).

The studies reporting on social inequalities in influenza outcomes in 1918 and in 2009, identified in this review, and also early research on social disparities in COVID-19 outcomes, often lacked a discussion of the possible mechanisms for the estimated social disparities, a framework to discuss those mechanisms and/or the data to separate the distal (social and policy) and proximal (behavioral and biological factors) factors for unequal exposure, susceptibility and access to health care leading to socially unequal pandemic outcomes [68]. Socially unequal exposure may relate to hand washing behavior or mask use, cleaning of surfaces, cramped living conditions, multigenerational living, occupational exposure, ability to work from home or stay away from work to care for family members, and use of public transportation. Social disparities in susceptibility may relate to poor nutritional status or, concurrent illnesses (e.g. NCDs). Finally, socioeconomic inequalities in understanding of or access to health advice (e.g. hand hygiene, social distancing, travel advisories) and vaccination or other public recommendations due to poor reading and writing skills may also explain part of the variation in outcomes by SES [13, 18].

Two of the studies on the 2009 pandemic included in our review, on Iran [35] and USA [55], reported increased risks (of infection rates in Iran and ICU stay vs. hospitalized non-ICU patients in USA) for those with high socioeconomic status–contrary to the authors' and our hypothesis. For the US study, the authors suggest that this may reflect social gradients in testing and demand for treatment and health care resources. In one matched case–control study of mortality among hospitalized H1N1 patients vs. H1N1 outpatients, no differences in any of the SES variables were found when controlling for health seeking behavior and barriers to

health care access. However, it is not clear whether these or other controls (age, sex, race, urban-rural, vaccination status, health behaviors, pre-existing conditions) "explained away" the negative associations of having health care insurance or the positive association of poverty in the univariate models. If the studies with data generated from health care systems that found higher pandemic risks for lower SES groups had controlled for a social gradient in testing and demand for treatment and health care resources, we expect that the findings of a social pattern in disease burden would be reinforced.

An important strength of our study is the use of a pre-registered study protocol for data gathering and analysis, which was peer-reviewed and published prior to the gathering of study data [22]. This helped ensure that the process was specified in a reproducible way and followed a rigorous and systematic workflow to identify studies and describe and analyze results. The engagement of professional information specialists to design, test and improve the literature search strategies that were applied to a broad range of literature databases is particularly important, given the lack of any previous systematic reviews on this topic with which our list of included studies could be compared.

Our study also has some potential limitations. First, we carried out our library search on 17 November 2017, and potential studies published in the two pre-COVID-19 years of 2018–19 and during the COVID-19 pandemic (2020–21) are not included. Given the massive research on COVID-19 pandemic in 2020–21 and the fact that the identified 2009 pandemic studies in our review were published rapidly after 2009–10, with 2012 being the average publication year, it is likely that we have missed a few 2009 pandemic studies. Given the strength and consistency of the results, we do not expect that newer studies would alter our general conclusions, at least not for the 2009 pandemic that was the topic of 35 of the 44 included studies. Second, we would note that the generalizability of our results is necessarily limited by the geographic focus of the research we synthesize: no studies using data from Africa were found, and few from Asia and South America. It is therefore reasonable to ask whether our results are representative outside of high-income countries in North America, Europe and the Oceania region.

## Conclusion

We have shown that influenza pandemic outcomes in 1918 and 2009 were associated with lower socioeconomic status and that pandemic outcomes in 2009 were not always socially neutral «great equalizers» once adjusting for medical risk factors [34, 46–48, 51, 54, 57, 59]. This resembles the finding of a study of COVID-19 hospital deaths demonstrating that medical risk factors did little to explain the higher risks of the deprived and of immigrants in the UK [69]. The social lessons from historical influenza pandemics such as those in 1918 or 2009 have not yet been taken into account in influenza pandemic preparedness [18], and this blind spot has also been evident in the response to the COVID-19 pandemic. Such social and ethnic vulnerability factors should be explicitly included and addressed in current and future plans and responses in order to more effectively reduce pandemic burdens, reduce social disparities and ameliorate the social consequences of future pandemics [70]. The global health and economic crisis created by the COVID-19 pandemic has made us only too aware of the need for a more holistic and comprehensive approach towards pandemic preparedness.

## Supporting information

**S1 Checklist. PRISMA 2020 checklist.**
(PDF)

**S1 File. PRISMA flow diagram.**
(PDF)

**S2 File. Specific studies included and all judgments and adjustments concerning inclusion and adjustments of reported numbers.**
(PDF)

**S1 Table. Medline search strategy.**
(PDF)

**S2 Table. Quality assessments.**
(PDF)

## Acknowledgments

We are indebted to our librarians Bettina Grødem Knutsen, Ingjerd Legreid Ødemark and Elisabeth Karlsen at the Learning Center and Library, Oslo Metropolitan University. Without their expertise and assistance in doing library searches this research would never have been accomplished.

## Author Contributions

**Conceptualization:** Svenn-Erik Mamelund, Clare Shelley-Egan.

**Data curation:** Svenn-Erik Mamelund, Clare Shelley-Egan, Ole Rogeberg.

**Formal analysis:** Ole Rogeberg.

**Funding acquisition:** Svenn-Erik Mamelund, Clare Shelley-Egan.

**Investigation:** Svenn-Erik Mamelund, Clare Shelley-Egan, Ole Rogeberg.

**Methodology:** Svenn-Erik Mamelund, Clare Shelley-Egan, Ole Rogeberg.

**Project administration:** Svenn-Erik Mamelund.

**Resources:** Svenn-Erik Mamelund, Clare Shelley-Egan, Ole Rogeberg.

**Software:** Ole Rogeberg.

**Supervision:** Svenn-Erik Mamelund, Ole Rogeberg.

**Validation:** Svenn-Erik Mamelund, Clare Shelley-Egan, Ole Rogeberg.

**Visualization:** Svenn-Erik Mamelund, Ole Rogeberg.

**Writing – original draft:** Svenn-Erik Mamelund, Clare Shelley-Egan, Ole Rogeberg.

**Writing – review & editing:** Svenn-Erik Mamelund, Clare Shelley-Egan, Ole Rogeberg.

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
