## [Decision Letter · Decision Letter 0]

29 Jun 2021

PONE-D-20-38870

The association between socioeconomic status and pandemic influenza: systematic review and meta-analysis

PLOS ONE

Dear Dr. Mamelund,

Thank you for submitting your manuscript to PLOS ONE. After careful consideration, we feel that it has merit but does not fully meet PLOS ONE’s publication criteria as it currently stands. Therefore, we invite you to submit a revised version of the manuscript that addresses the points raised during the review process.

We look forward to receiving your revised manuscript.

Kind regards,

Obinna Ikechukwu Ekwunife, PhD

Academic Editor

PLOS ONE

Journal Requirements:

2.  Our staff editors have determined that your manuscript is likely within the scope of our Call for Papers on Influenza. This editorial initiative is headed by PLOS ONE Guest Editors Dr. Meagan Deming and Dr. Deshayne Fell. The Collection encompasses research on influenza prevention on every level, including in vitro, translational, behavioral, and clinical studies; disease and immunity modelling; as well as new approaches to influenza prevention. Additional information can be found on our announcement page: https://collections.plos.org/call-for-papers/influenza/.

Currently, your manuscript is included in the group of papers being considered for this call. Please note that being considered for the Collection does not require additional peer review beyond the journal’s standard process and will not delay the publication of your manuscript if it is accepted by PLOS ONE. We would greatly appreciate your confirmation that you would like your manuscript to be considered for this Collection by indicating this in your next cover letter. If you would prefer to remove your manuscript from collection consideration, please specify this in your cover letter.

If you would like your manuscript to be included in the call for papers, please expand upon your discussion of how your results relate to influenza prevention and/or preparedness in your introduction and/or discussion, as this is the main focus of this call.

3. We note that this manuscript is a systematic review or meta-analysis; our author guidelines therefore require that you use PRISMA guidance to help improve reporting quality of this type of study. Please upload copies of the completed PRISMA checklist as Supporting Information with a file name “PRISMA checklist”.

Reviewers' comments:

Reviewer's Responses to Questions

**Comments to the Author**

1. Is the manuscript technically sound, and do the data support the conclusions?

Reviewer #1: Yes

Reviewer #2: No

2. Has the statistical analysis been performed appropriately and rigorously? 

Reviewer #1: Yes

Reviewer #2: No

3. Have the authors made all data underlying the findings in their manuscript fully available?

Reviewer #1: Yes

Reviewer #2: Yes

4. Is the manuscript presented in an intelligible fashion and written in standard English?

Reviewer #1: Yes

Reviewer #2: No

5. Review Comments to the Author

Reviewer #1: This manuscript is a review and metanalysis of papers that study the relationship between socioeconomic status and pandemic influenza. They find evidence across these works that there is an association between lower SES and increased influenza, and that this relationship is independent of underlying medical conditions. The authors took a thorough approach and explain the work clearly.

Major comments

The authors did not include race as one of their measures of SES, and did not include race as one of the keywords in their search. This likely makes sense due to the global nature of the reviewed papers and different relationships between race and SES across different countries, but the authors should justify this choice. For example, a fair number of the papers were from the US, where race and SES are highly correlated, and much work may have been done relating race and infectious disease that is being overlooked.

The authors perhaps over-explain their statistical methods, including a lot of justification and description. While this is not necessarily a problem, the paper and results could be more succinct if the methods and results sections were honed to only include the necessary information, removing additional details that readers familiar with the methods would already know, or could find out from reading the basics of the mentioned statistical method.

The authors do not pay attention to healthcare utilization or healthcare access differences in the papers analyzed, though most of the papers include data that is generated from healthcare systems. Different levels of healthcare access or different trends in healthcare utilization are likely to be present across SES levels, biasing who the data captures and the results. It would be useful if the authors could describe whether any of the papers analyzed accounted for healthcare utilization in any way, and if this plays a role.

Minor comments

The sentence on line 48 is confusing “The studies typically lacked…”

In lines 335-338, “a…risks” is repeated several times and is grammatically incorrect

Reviewer #2: Thank you for the opportunity to review this paper. I find the objective interesting, but it appears that this study has no significant addition to knowledge. Only 1918 and 2009 pandemics were included. Besides, a systematic review for 2009 pandemic has been published, and I find 1918 pandemic as obsolete to judge what happens in today’s global health system. There is a higher inequality and disparity in societies 100 years ago compared to 2009. So, a pooled effect of 1918 and 2009 possess some validity issues. Also, given the very high heterogeneity, the validity of the meta-analysis is questionable.

Generally, the presentation of this paper needs to be improved in terms of language, choice of words, punctuations, and special characters.

Here are some specific points below:

Abstract

1. Please check for use of past and present tense all through the paper.

E.g., “Results are summarized narratively and using meta-analytic strategies”.

2. “Studies comparing severe outcomes (ICU or death) to hospital admissions are few but indicate no clear association.”

“..Indicate no clear association” or “indicate no significant association”?

The term “clear” does not sound statistically right.

3. The conclusion was poorly written. They conclusion should be clear with a concise take-home message linked to the study objective – “whether and to what extent there was an

association between socioeconomic status (SES) and disease outcomes in the last five

influenza pandemics”. See the authors conclusion below:

“Results show that social risk factors help to explain pandemic outcomes in 1918 and in 2009 although the mechanisms and types of social vulnerabilities leading to disparities in outcomes may differ over time. Studies of the 2009 pandemic also showed that social vulnerability could not always be explained by medical risk factors.”

This should be in the discussion section of this paper. Also, there should be a full stop mark after 2009.

A good example of conclusion should be: “Socioeconomic status are not associated with pandemic influenza”. OR “There is no significant association between socioeconomic status and pandemic influenza”.

4. What is the significance level of the pooled estimate? This is important for a meta-analysis result but found to be lacking in the abstract.

Introduction

1. Page 3, Line 54 – 56: “A systematic assessment of the evidence for such risk inequalities has been lacking, however, apart from a systematic review and meta-analysis of how the risk of 2009 influenza pandemic outcomes differ for disadvantaged populations (mainly indigenous people) (20).”

I am not fully convinced of the addition to knowledge of this study. The authors already stated in the abstract that they only got data for the 1918 and 2009 pandemics. 1918 seems too obsolete (over a 100 years), which can affect the internal validity of their findings. Already, a metanalysis for the 2009 pandemic has been performed, so the addition of 1918 pandemic association with SES (which appears obsolete to this reviewer), questions the addition to knowledge of this paper.

Second, the above statement still needs to be rephrased: E.g., “apart from a systematic review and meta-analysis of how the risk of 2009 influenza pandemic outcomes differ for disadvantaged populations (mainly indigenous people) (20), a systematic assessment of the evidence for such risk inequalities has been lacking.

In summary of this point, I find no substantial addition to knowledge of this paper, in its current state.

2. Page 4, line 69 - 74: “Our review identified studies on the 1918 and 2009 pandemics only, with evidence of a social gradient in the disease burden of both these pandemics……….”.

It is wrong to place this statement in the introduction. The last paragraph of the introduction should be in the results section.

3. Materials and methods:

I found the search period to be obsolete. 2017 is not a recent search. What if recent studies in 2019 or 2020 exist that collected retrospective data from hospitals/agencies, and performed studies on the association of SES and influenza? How are the authors sure, some studies were not missed? I expect to see search up to at least June 2020.

4. Page 5, line 92: “However, we did not get any responses that made it into the paper and our analysis”

Authors should check for word choice here.

5. The inclusion and exclusion criteria are too lengthy. The authors mixed up inclusion criteria and search strategy. This needs to be clearly distinguished for ease of reading.

6. Page 6, line 135: “pilot tested”…

There should be a hyphen

7. Data synthesis is deficient. The content is poorly arranged. Some information does not need to be in this section but previous sections. Also, the authors should use reported speech. The writing style seems like a proposal of what they plan to do, when they have already completed the study.

The authors need to improve on their description of this section. It is structurally poor.

8. Page 37, line 373: Quantitative meta-analysis

All meta-analyses are quantitative. Better to say, “Quantitative synthesis”.

No information about the degree of heterogeneity that should prevent quantitative synthesis.

6. PLOS authors have the option to publish the peer review history of their article (what does this mean?). If published, this will include your full peer review and any attached files.

Reviewer #1: No

Reviewer #2: **Yes: **Charles Okafor

---

## [Author Response · Author response to Decision Letter 0]

9 Jul 2021

Reviewer #1: This manuscript is a review and metanalysis of papers that study the relationship between socioeconomic status and pandemic influenza. They find evidence across these works that there is an association between lower SES and increased influenza, and that this relationship is independent of underlying medical conditions. The authors took a thorough approach and explain the work clearly.

-Response from authors: Thanks so much.

Major comments

The authors did not include race as one of their measures of SES, and did not include race as one of the keywords in their search. This likely makes sense due to the global nature of the reviewed papers and different relationships between race and SES across different countries, but the authors should justify this choice. For example, a fair number of the papers were from the US, where race and SES are highly correlated, and much work may have been done relating race and infectious disease that is being overlooked.

-Response from authors: This paper did not do a review of the association between race, ethnicity, and indigenous people status or “disadvantaged” populations. It rather looked at socioeconomic status (SES) indices at individual or aggregate level such as income, education and summary measures of SES. However, as stated in the “Inclusion criteria for title and abstract screening”-section in the original submitted version (page 6, lines 113-115, “The search strategy also covered studies of ethnic and disadvantaged populations, as some of these included covariates for socioeconomic confounders”. To make it clear that these papers were included, in the revised version of “Inclusion criteria for title and abstract screening”-section, we included a new bullet point stating (page 5, line 110): “Studies of race, ethnicity, and indigenous people that reported data on SES controls”.

The authors perhaps over-explain their statistical methods, including a lot of justification and description. While this is not necessarily a problem, the paper and results could be more succinct if the methods and results sections were honed to only include the necessary information, removing additional details that readers familiar with the methods would already know, or could find out from reading the basics of the mentioned statistical method.

-Response from authors: A meta-analysis summarizes the results of a field of research, but many researchers in that field may lack expertise in meta-analytic methods. We believe it is useful to present some intuition/explanation that makes the main assumptions of different meta-analytic models more comprehensible to researchers, but recognize that there is a trade-off in that this quickly feels excessive and unnecessary to researchers already familiar with such tools. We have now revised the “Data synthesis” section to substantially shorten the explanations – while still making the section understandable to those without prior knowledge of meta-analytic techniques.

The authors do not pay attention to healthcare utilization or healthcare access differences in the papers analyzed, though most of the papers include data that is generated from healthcare systems. Different levels of healthcare access or different trends in healthcare utilization are likely to be present across SES levels, biasing who the data captures and the results. It would be useful if the authors could describe whether any of the papers analyzed accounted for healthcare utilization in any way, and if this plays a role.

-Response from authors: Thanks, this is very good comment. We do not study health care utilization as an outcome, but pandemic disease burden. However, already in the originally submitted version we included a couple of sentences on this issue (page 42, lines 511-514). In our revisions, we have now added a new sentences on this topic on page 41, lines 498-510.

Minor comments

The sentence on line 48 is confusing “The studies typically lacked…”

-Response from authors: We changed this sentence to: “However, these studies used aggregate-level data, were mainly descriptive and did not use multivariate statistical models”.

In lines 335-338, “a…risks” is repeated several times and is grammatically incorrect

-Response from authors: Thanks – see our revisons.

Reviewer #2: Thank you for the opportunity to review this paper. I find the objective interesting, but it appears that this study has no significant addition to knowledge. Only 1918 and 2009 pandemics were included. Besides, a systematic review for 2009 pandemic has been published, and Generally, the presentation of this paper needs to be improved in terms of language, choice of words, punctuations, and special characters.

[…] I find 1918 pandemic as obsolete to judge what happens in today’s global health system. There is a higher inequality and disparity in societies 100 years ago compared to 2009. So, a pooled effect of 1918 and 2009 possess some validity issues. Also, given the very high heterogeneity, the validity of the meta-analysis is questionable.

-Authors response: We agree that there are good reasons to suspect that socio-economic disparities could have different implications for a flu pandemic in 2009 and 1918, perhaps particularly if we consider comparisons within Western countries. When we perform the subsample analysis shown in table 2, however, the pooled effect mean is nonetheless highly similar for the 1918 subsample (1.4, with 95% CI 1.1-1.8) and the 2009 subsample (1.4, CI: 1.2-1.8). No indication of substantive differences between the two periods is found in the Bayesian model simultaneously accounting for multiple study-level covariates either. This was surprising to us – like the reviewer we expected larger associations in the 1918 studies.

-Authors response: We agree that there is high heterogeneity across studies, as we discuss on page 36, but we do not agree that this in itself means that the meta-analysis is invalid. The random effect model is built on the assumption that there is (potentially substantial) study level effect differences, and so heterogeneity in itself should not invalidate the results – although it will substantially affect the interpretation of the results. In particular, when there is high heterogeneity the pooled effect mean will not be very representative of the “typical effect” underlying individual studies. We already note this in the discussion on page 36, where we write that about 50% of study-level effects would be expected to fall in the range of 1.1 to 1.9 given the estimates from the random effect model. This heterogeneity was not explicitly noted in the discussion section, however, which we have now done.

Here are some specific points below:

Abstract

1. Please check for use of past and present tense all through the paper.

E.g., “Results are summarized narratively and using meta-analytic strategies”.

2. “Studies comparing severe outcomes (ICU or death) to hospital admissions are few but indicate no clear association.”

“..Indicate no clear association” or “indicate no significant association”?

The term “clear” does not sound statistically right.

-Authors response: Thanks for these comments. We have now changed the use of past to present tense in the abstract and throughout the paper. We have also made it clear what findings that were statistically significant.

3. The conclusion was poorly written. They conclusion should be clear with a concise take-home message linked to the study objective – “whether and to what extent there was an

association between socioeconomic status (SES) and disease outcomes in the last five

influenza pandemics”. See the authors conclusion below:

“Results show that social risk factors help to explain pandemic outcomes in 1918 and in 2009 although the mechanisms and types of social vulnerabilities leading to disparities in outcomes may differ over time. Studies of the 2009 pandemic also showed that social vulnerability could not always be explained by medical risk factors.”

This should be in the discussion section of this paper. 

Also, there should be a full stop mark after 2009.

A good example of conclusion should be: “Socioeconomic status are not associated with pandemic influenza”. OR “There is no significant association between socioeconomic status and pandemic influenza”.

-Authors response: Thanks for your suggestions. The conclusion in the abstract now reads: “We found that SES was significantly associated with pandemic influenza outcomes with people of lower SES having the highest disease burden in both 1918 and 2009. To prepare for future pandemics, we must consider social vulnerability”.

4. What is the significance level of the pooled estimate? This is important for a meta-analysis result but found to be lacking in the abstract.

-Authors response: We have now added the p-value in the abstract: “….we found a pooled mean odds ratio of (1.4 (95% CI: 1.2 – 1.7, p < 0.001),…”

Introduction

1. Page 3, Line 54 – 56: “A systematic assessment of the evidence for such risk inequalities has been lacking, however, apart from a systematic review and meta-analysis of how the risk of 2009 influenza pandemic outcomes differ for disadvantaged populations (mainly indigenous people) (20).”

I am not fully convinced of the addition to knowledge of this study. The authors already stated in the abstract that they only got data for the 1918 and 2009 pandemics. 1918 seems too obsolete (over a 100 years), which can affect the internal validity of their findings. Already, a metanalysis for the 2009 pandemic has been performed, so the addition of 1918 pandemic association with SES (which appears obsolete to this reviewer), questions the addition to knowledge of this paper.

Second, the above statement still needs to be rephrased: E.g., “apart from a systematic review and meta-analysis of how the risk of 2009 influenza pandemic outcomes differ for disadvantaged populations (mainly indigenous people) (20), a systematic assessment of the evidence for such risk inequalities has been lacking.

In summary of this point, I find no substantial addition to knowledge of this paper, in its current state.

-Authors response: To make it clearer how our study is contributing we have added this sentence: “Apart from a systematic review and meta-analysis of the 2009 pandemic disease burden in low and low to middle income economies and differences in disease outcomes in that pandemic for ethnic minorities vs non-ethnic minorities (20), a systematic assessment of several historical influenza pandemics and of the evidence for disparities in pandemic outcomes by individual and/or area-level SES (education, income, household crowding and quality, unemployment, occupation-based social class, poverty status, share below poverty levels, deprivation indexes etc.) has been lacking”

2. Page 4, line 69 - 74: “Our review identified studies on the 1918 and 2009 pandemics only, with evidence of a social gradient in the disease burden of both these pandemics……….”.

It is wrong to place this statement in the introduction. The last paragraph of the introduction should be in the results section.

-Authors response: We have now deleted this section. 

3. Materials and methods:

I found the search period to be obsolete. 2017 is not a recent search. What if recent studies in 2019 or 2020 exist that collected retrospective data from hospitals/agencies, and performed studies on the association of SES and influenza? How are the authors sure, some studies were not missed? I expect to see search up to at least June 2020.

-Authors response: We added some revisions in the limitation-section of the paper: “Our study also has some potential limitations. First, we carried out our library search on 17 November 2017, and potential studies published in the two pre-COVID-19 years of 2018-19 and during the COVID-19 pandemic (2020-21) are not included. Given the massive research on COVID-19 pandemic in 2020-21 and the fact that the identified 2009 pandemic studies in our review were published rapidly after 2009-10, with 2012 being the average publication year, it is likely that we have missed a few 2009 pandemic studies. Given the strength and consistency of the results, we do not expect that newer studies would alter our general conclusions, at least not for the 2009 pandemic that was the topic of 35 of the 44 included studies”.

4. Page 5, line 92: “However, we did not get any responses that made it into the paper and our analysis”

Authors should check for word choice here.

-Authors response: We changed the wordings to “However, we did not get any responses to these requests”.

5. The inclusion and exclusion criteria are too lengthy. The authors mixed up inclusion criteria and search strategy. This needs to be clearly distinguished for ease of reading.

-Authors response: Thanks for seeing this. We removed a paragraph on keywords for SES, morbidity and mortality from the section on inclusion criteria to the section on search strategy. We also shortened the inclusion and exclusion criteria sections. 

6. Page 6, line 135: “pilot tested”…

There should be a hyphen

-Authors response: Thanks. Changed.

7. Data synthesis is deficient. The content is poorly arranged. Some information does not need to be in this section but previous sections. Also, the authors should use reported speech. The writing style seems like a proposal of what they plan to do, when they have already completed the study.

The authors need to improve on their description of this section. It is structurally poor.

-Authors response: We have changed wording to reported speech and also made numerous revisions in the section on “Data synthesis”.

8. Page 37, line 373: Quantitative meta-analysis

All meta-analyses are quantitative. Better to say, “Quantitative synthesis”.

-Authors response: Thanks. Changed 

No information about the degree of heterogeneity that should prevent quantitative synthesis.

-Authors response: While we agree that there is substantial effect heterogeneity across the included studies, we do not believe this prevents a quantitative synthesis. As we write, such variation should be expected given the variation in outcome measures, indicators, region, period etc (see second paragraph of the Data Synthesis section on page 9), but it may still be possible to use quantitative techniques to characterize the distribution of effects underlying these studies – as well as assess whether the variation is strongly related to study level covariates. The important thing is that this underlying variation is clearly noted and its implications for the interpretation noted and taken into account, which we do when presenting the results on page 36.

---

## [Decision Letter · Decision Letter 1]

13 Aug 2021

The association between socioeconomic status and pandemic influenza: systematic review and meta-analysis

PONE-D-20-38870R1

Dear Dr. Svenn-Erik Mamelund, 

We’re pleased to inform you that your manuscript has been judged scientifically suitable for publication and will be formally accepted for publication once it meets all outstanding technical requirements.

Kind regards,

Obinna Ikechukwu Ekwunife, PhD

Academic Editor

PLOS ONE

Additional Editor Comments (optional):

Reviewers' comments:

Reviewer's Responses to Questions

**Comments to the Author**

1. If the authors have adequately addressed your comments raised in a previous round of review and you feel that this manuscript is now acceptable for publication, you may indicate that here to bypass the “Comments to the Author” section, enter your conflict of interest statement in the “Confidential to Editor” section, and submit your "Accept" recommendation.

Reviewer #2: All comments have been addressed

2. Is the manuscript technically sound, and do the data support the conclusions?

Reviewer #2: Yes

3. Has the statistical analysis been performed appropriately and rigorously? 

Reviewer #2: Yes

4. Have the authors made all data underlying the findings in their manuscript fully available?

Reviewer #2: Yes

5. Is the manuscript presented in an intelligible fashion and written in standard English?

Reviewer #2: Yes

6. Review Comments to the Author

Reviewer #2: The authors have addressed my concerns. The mnuascript is sound and the statisitcis is okay. The editor should consider it for publication.

7. PLOS authors have the option to publish the peer review history of their article (what does this mean?). If published, this will include your full peer review and any attached files.

Reviewer #2: **Yes: **Charles Ebuka Okafor

---

## [Editor Report · Acceptance letter]

25 Aug 2021

PONE-D-20-38870R1 

The association between socioeconomic status and pandemic influenza: systematic review and meta-analysis 

Dear Dr. Mamelund:

I'm pleased to inform you that your manuscript has been deemed suitable for publication in PLOS ONE. Congratulations! Your manuscript is now with our production department. 

Kind regards, 

on behalf of

Dr. Obinna Ikechukwu Ekwunife 

Academic Editor

PLOS ONE